JCB Journal of Cell Biology

# Nedd4-2-dependent regulation of astrocytic Kir4.1 and Connexin43 controls neuronal network activity

Bekir Altas[1,2,3,4], Hong-Jun Rhee[1]*, Anes Ju[1,3]*, Hugo Cruces Solís[1,2,5]*, Samir Karaca[2,6]*, Jan Winchenbach[3,5], Oykum Kaplan-Arabaci[1,7], Manuela Schwark[1], Mateusz C. Ambrozkiewicz[1,2,8], ChungKu Lee[1], Lena Spieth[5], Georg L. Wieser[9], Viduth K. Chaugule[10], Irina Majoul[11], Mohamed A. Hassan[1,12], Rashi Goel[1], Sonja M. Wojcik[1], Noriko Koganezawa[13], Kenji Hanamura[13], Daniela Rotin[14], Andrea Pichler[10], Miso Mitkovski[9], Livia de Hoz[5], Alexandros Poulopoulos[4], Henning Urlaub[6,15,16], Olaf Jahn[17,18], Gesine Saher[5], Nils Brose[1], JeongSeop Rhee[1], and Hiroshi Kawabe[1,13,19,20]

Nedd4-2 is an E3 ubiquitin ligase in which missense mutation is related to familial epilepsy, indicating its critical role in regulating neuronal network activity. However, Nedd4-2 substrates involved in neuronal network function have yet to be identified. Using mouse lines lacking Nedd4-1 and Nedd4-2, we identified astrocytic channel proteins inwardly rectifying K⁺ channel 4.1 (Kir4.1) and Connexin43 as Nedd4-2 substrates. We found that the expression of Kir4.1 and Connexin43 is increased upon conditional deletion of Nedd4-2 in astrocytes, leading to an elevation of astrocytic membrane ion permeability and gap junction activity, with a consequent reduction of γ-oscillatory neuronal network activity. Interestingly, our biochemical data demonstrate that missense mutations found in familial epileptic patients produce gain-of-function of the *Nedd4-2* gene product. Our data reveal a process of coordinated astrocytic ion channel proteostasis that controls astrocyte function and astrocyte-dependent neuronal network activity and elucidate a potential mechanism by which aberrant Nedd4-2 function leads to epilepsy.

## Introduction

Neuronal network activity depends critically on the properties of the nerve cells involved—e.g., synaptic connectivity, synapse strength, and excitation–inhibition balance. In addition, glia cells play important roles as modulators of neuronal and circuit function. For instance, gliotransmitters released from astrocytes control neuronal functions by binding to NMDA receptors (Araque et al., 1998; Clasadonte et al., 2013), and the release of S100β induces γ-oscillations (Sakatani et al., 2008). Astrocytes act as important regulators of ion homeostasis in the brain via astrocytic ion channels (e.g., inwardly rectifying K⁺ channels, Kir4.1 and Kir5.1), ion pumps (e.g., Na⁺-K⁺ ATPase), and gap junctions, all of which play roles in ion and solute exchange with the extracellular space and between adhering astrocytes (Chever et al., 2010; Haglund and Schwartzkroin, 1990; Takumi et al., 1995; Wallraff et al., 2006), thus acutely regulating the extracellular environment that determines neuronal excitability.

The most prominent K⁺-permeable astrocytic channel is Kir4.1, which plays a key role in the clearance of excess

[1]Department of Molecular Neurobiology, Max Planck Institute for Multidisciplinary Sciences, Göttingen, Germany; [2]International Max Planck Research School and the Göttingen Graduate School for Neurosciences, Biophysics and Molecular Biosciences, Göttingen, Germany; [3]The Göttingen Graduate School for Neurosciences, Biophysics, and Molecular Biosciences, PhD Program Systems Neuroscience, University of Göttingen, Göttingen, Germany; [4]Department of Pharmacology and Program in Neuroscience, University of Maryland School of Medicine, Baltimore, MD, USA; [5]Department of Neurogenetics, Max Planck Institute for Multidisciplinary Sciences, Göttingen, Germany; [6]Bioanalytical Mass Spectrometry Group, Max Planck Institute for Multidisciplinary Sciences, Göttingen, Germany; [7]The Göttingen Graduate School for Neurosciences, Biophysics, and Molecular Biosciences, PhD Program Molecular Physiology of the Brain, University of Göttingen, Göttingen, Germany; [8]Institute of Cell Biology and Neurobiology, Charité-Universitätsmedizin Berlin, Corporate Member of Freie Universität Berlin, Humboldt-Universität zu Berlin, and Berlin Institute of Health, Berlin, Germany; [9]City Campus Light Microscopy Facility, Max Planck Institute for Multidisciplinary Sciences, Göttingen, Germany; [10]Department of Epigenetics, Max Planck Institute of Immunobiology and Epigenetics, Freiburg, Germany; [11]Institute of Biology, Center for Structural and Cell Biology in Medicine, University of Lübeck, Lübeck, Germany; [12]Protein Research Department, Genetic Engineering and Biotechnology Research Institute (GEBRI), City of Scientific Research and Technological Applications (SRTA-City), New Borg El-Arab City, Egypt; [13]Department of Pharmacology, Gunma University Graduate School of Medicine, Maebashi, Japan; [14]The Hospital for Sick Children and University of Toronto, Toronto, Canada; [15]Bioanalytics, Institute for Clinical Chemistry, University Medical Center Göttingen, Göttingen, Germany; [16]Cluster of Excellence "Multiscale Bioimaging: From Molecular Machines to Networks of Excitable Cells" (MBExC), University of Göttingen, Göttingen, Germany; [17]Department of Molecular Neurobiology, Neuroproteomics Group, Max Planck Institute for Multidisciplinary Sciences, Göttingen, Germany; [18]Department of Psychiatry and Psychotherapy, Translational Neuroproteomics Group, University Medical Center Göttingen, Göttingen, Germany; [19]Division of Pathogenetic Signaling, Department of Biochemistry and Molecular Biology, Kobe University Graduate School of Medicine, Kobe, Japan; [20]Department of Gerontology, Laboratory of Molecular Life Science, Institute of Biomedical Research and Innovation, Foundation for Biomedical Research and Innovation at Kobe, Kobe, Japan.

*H.-J. Rhee, A. Ju, H.C. Solís, and S. Karaca contributed equally to this paper.   Correspondence to Bekir Altas: altas@em.mpg.de;   JeongSeop Rhee: rhee@em.mpg.de; Hiroshi Kawabe: kawabe@gunma-u.ac.jp.

extracellular K⁺. Its brain-specific conditional knock-out (KO), i.e., *Kir4.1*^f/f^;GFAP-Cre in mice causes a reduction of electrophysiologically measured whole-cell ion currents in astrocytes, and KO animals show pronounced body tremor, ataxia, and tonic-clonic seizures (Djukic et al., 2007). Similar symptoms are seen in patients with SeSAME- or EAST-syndrome, who carry frame-shift or missense mutations in the coding sequence of *KCNJ10*, the human ortholog of the *Kir4.1* gene (Bockenhauer et al., 2009; Scholl et al., 2009). Extracellular K⁺ taken up by astrocytes is quickly dissipated within an astrocytic network connected by gap junctions, which are mainly composed of Connexin43 (Cx43) and Connexin30 (Cx30) and are required for the dampening of neuronal network activity (Wallraff et al., 2006).

The cell surface expression and turnover of ion channels are regulated by protein ubiquitination (Kimura et al., 2011; Staub and Rotin, 2006). It is mediated by the successive action of ubiquitin (Ub) activating enzymes (E1), Ub conjugating enzymes (E2), and Ub ligases (E3), of which the E3s define substrate specificity (Hershko and Ciechanover, 1998). Ub itself has seven lysine residues that can act as Ub acceptors so that various types of polyUb chains can be formed. The most widely known K48-linked polyUb chains target substrate proteins to proteasomal degradation, while plasma membrane proteins with K63-linked polyUb chains are endocytosed and lysosomally degraded (Piper et al., 2014). Neuronal precursor–expressed developmentally downregulated 4 (Nedd4) family E3s have been implicated in monoubiquitination and K63-linked polyUb chain formation (Kawabe et al., 2010; Maspero et al., 2013; Woelk et al., 2006). They belong to the homologous E6AP carboxyl terminus (HECT)-type E3 superfamily. Among the Nedd4 family E3s, Nedd4-2 (N4-2) is of particular interest because missense mutations of evolutionarily conserved residues (S233L, E271A, and H515P) were found in patients with familial photosensitive epilepsy (Dibbens et al., 2007; Vanli-Yavuz et al., 2015) and several other de novo mutations in N4-2 were reported in epilepsy patients (Allen et al., 2013), indicating that N4-2 is a critical regulator of neuronal network activity.

In the present study, we identified Kir4.1 and Cx43 as the most prominent substrates of N4-2 in a quantitative mass spectrometry screen. We found that N4-2-deficient astrocytes exhibit elevated levels of Kir4.1 and Cx43, which cause increased plasma membrane ion permeability and gap junction coupling. Reflecting these cellular changes, astrocyte-specific N4-2 KO mice show severely altered neuronal network synchronicity as indicated by a strong reduction of the power of γ-oscillatory activity in the brain. This deficiency is restored by the application of pharmacological blockers of Kir4.1 or Cx43. Our data unravel a novel regulatory principle by which a single E3 ligase N4-2 coordinately controls the surface expression of two ion channels critical for fundamental aspects of astrocyte function, i.e., the plasma membrane ion permeability and gap junction network connectivity, which, in turn, are key determinants of neuronal network synchronicity. Additionally, we determine that pathological N4-2 point mutations found in familial photosensitive epilepsy patients are gain-of-function mutations, indicating that inappropriate enhancement of N4-2-dependent ubiquitination and degradation of Kir4.1 and Cx43 could be a potential cause of familial epilepsy.

## Results

### N4-2 is a critical regulator of neuronal network synchronicity

Missense mutations of evolutionarily conserved residues of N4-2 (S233L, E271A, and H515P) were found in patients with familial photosensitive epilepsy (Dibbens et al., 2007; Vanli-Yavuz et al., 2015). Given that enhanced γ-band oscillatory activities could be one of the potential triggers of photosensitive epilepsy (Parra et al., 2003; Perry et al., 2014), we studied the impact of the loss of N4-2 alone or together with its closest isoform Nedd4-1 (N4-1) on γ-oscillations in hippocampal CA3 region of acute brain slices. γ-Oscillations were induced by 100 nM kainate and recorded as local field potentials (LFPs). The average power of γ-oscillatory activity was significantly reduced in brain-specific conditional N4-1/N4-2 KO mice (*Nedd4-1*^f/f^;*Nedd4-2*^f/f^; *EMX*-Cre; N4-1/2 bDKO; Gorski et al., 2002; Kawabe et al., 2010; Kimura et al., 2011) or showed a strong trend toward a significant reduction (P = 0.076) in brain-specific N4-2 KO mice (*Nedd4-2*^f/f^;*EMX*-Cre; N4-2 bKO) as compared with their corresponding controls (*Nedd4-1*^f/f^;*Nedd4-2*^f/f^; N4-1/2 CTL and *Nedd4-2*^f/f^; N4-2 CTL, respectively) when tested with a stringent statistical test (i.e., nested *t* test; Fig. 1). Student's *t* test indicated that N4-2 bKOs show a significantly reduced power of γ-oscillatory activity as compared with controls (P = 0.014). This phenotype was more pronounced in N4-1/2 bDKO than in N4-2 bKO, and neuron-specific N4-1/N4-2 KO mice (*Nedd4-1*^f/f^; *Nedd4-2*^f/f^;*Nex*-Cre; N4-1/2 nDKO; Goebbels et al., 2006) and N4-1 bKO did not show the phenotype (Fig. S1), indicating that non-neuronal N4-2 plays a prominent role in maintaining the neuronal network.

### Gain of E3 function by a missense mutation of N4-2 in epileptic patients

Amino acid residues mutated in N4-2 of familial photosensitive epilepsy patients (S233L, E271A, H515P) are not located within the substrate-recognizing WW domains (Dibbens et al., 2007; Vanli-Yavuz et al., 2015). We thus hypothesized that these mutations might affect the intrinsic enzymatic activity of N4-2 rather than substrate binding. To test this hypothesis, we purified recombinant WT and two mutant (S233L and H515P) N4-2 proteins and analyzed their enzymatic activities by in vitro ubiquitination assays with Ub as substrate (Fig. 2). In these experiments, the reduction of monomeric Ub (i.e., free Ub) and the formation of polyUb chains serve as readouts for N4-2 enzyme activity (Fig. 2, A and B). Recombinant N4-2 proteins with a missense mutation, S233L or H515P, were purified and used for the in vitro ubiquitination assays (Fig. 2 C). Despite several attempts, N4-2 with E271A mutation could not be purified. The depletion of monomeric Ub and formation of polyUb chains in this assay was significantly enhanced in mutant N4-2 within the linear range of the assay time course (Fig. 2, D–G). These results indicate that the individual S233L and H515P mutations cause a gain-of-function of N4-2 E3 activity, which might result in increased ubiquitination levels and reduced expression of

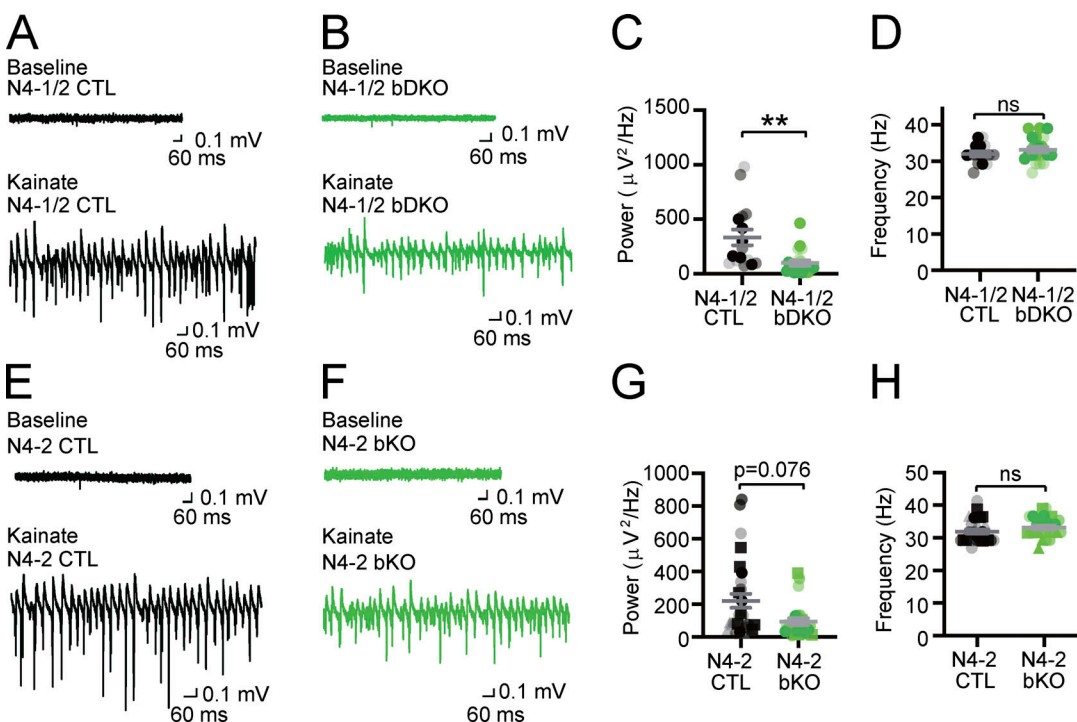

Figure 1. **γ-oscillations in acute slices from Nedd4 brain-specific knockout mice. (A–D)** Reduced power of γ-oscillations in *Nedd4-1*[f/f];*Nedd4-2*[f/f];*EMX*-Cre (N4-1/2 bDKO) mice. Representative recordings in CA3 hippocampal regions of acute brain slices from *Nedd4-1*[f/f];*Nedd4-2*[f/f] (N4-1/2 CTL) (A) and N4-1/2 bDKO (B) mice before (baseline) and during (Kainate) induction of γ-oscillations with 100 nM kainate. Average powers (C) and frequencies (D) of γ-oscillations in N4-1/2 CTL (black dots) and N4-1/2 bDKO (green dots). **(E–H)** Reduced power of γ-oscillations in *Nedd4-2*[f/f];*EMX*-Cre (N4-2 bKO) mice. Representative recordings in the CA3 region of hippocampus slices from *Nedd4-2*[f/f] (N4-2 CTL) (E) and N4-2 bKO (F) mice before (Baseline) and during (Kainate) the application of 100 nM kainate. Average powers (G) and frequencies (H) of γ-oscillations in N4-2 CTL (black dots) and N4-2 bKO (green dots). Different tones and shapes of dots in dot plots in C, D, G, and H represent data from different mice. Numbers of recorded slices ($n$) and the animal number ($N$); (C and D), $n = 16$ and $N = 3$ for N4-1/2 CTL, $n = 21$ and $N = 3$ for N4-1/2 bDKO; (G and H), $n = 29$ and $N = 6$ for N4-2 CTL, $n = 25$ and $N = 5$ for N4-2 bKO. Results are shown as mean ± SEM. **, $0.001 < P < 0.01$; *, $0.01 < P < 0.05$; ns, $0.05 < P$ (two-tailed nested $t$ test). Data distribution was assumed to be normal but this was not formally tested. See also Fig. S1 and Table S2.

substrates in patients' brains and in increased power of γ-oscillations, as reported in photosensitive epileptic patients (Perry et al., 2014).

## A screen for N4-1 and N4-2 substrates in the brain identifies Kir4.1 and Cx43

N4-1 and N4-2 generate K63-linked polyUb chains (Maspero et al., 2013), which are involved in several biological processes including the endocytosis and lysosomal degradation of transmembrane proteins (Acconcia et al., 2009). In contrast to a previous study (Zhu et al., 2017), the levels of the glutamate receptor subunit GluA1 were not changed in N4-1/2 bDKOs (Fig. 3, A–C). This result indicates that the reduced power of γ-oscillations in N4-1/2 bDKOs is not caused by aberrant total GluA1 level although the involvement of GluA2 cannot be excluded. Given the almost same expression levels of N4-2 in the cortex and hippocampus (Fig. 3 D), we purified synaptic plasma membrane fractions (SM) from N4-1/2 bDKO and N4-1/2 CTL cortices (Fig. 3 E; Mizoguchi et al., 1989) and compared protein profiles using isobaric tags for relative and absolute quantitation (iTRAQ; Ross et al., 2004; Schmidt et al., 2013). Intriguingly, two astrocytic membrane proteins, the K⁺ channel Kir4.1 and the gap junction alpha 1 protein (Gja1 or Cx43) were found as the two

most upregulated proteins in N4-1/2 bDKO samples (Fig. 3 F and Table S1). The results obtained by proteomic screening were validated by quantitative Western blotting for Kir4.1 and Cx43 in N4-1/2 bDKO and N4-1/2 CTL samples using the cortical lysates (Fig. 3, G–I). Levels of EAAT2, which has a subcellular distribution akin to Kir4.1 and Cx43, were not changed in N4-1/2 bDKO samples (Fig. 3, E, J, and K), demonstrating that the upregulation of Kir4.1 and Cx43 in N4-1/2 bDKO samples is not due to an increase in astrocyte numbers in N4-1/2 bDKO cortex. This notion was supported by quantitative immunohistochemistry, as shown in Fig. 3, L and M. The levels of Kir4.1 and Cx43 mRNAs were not different between N4-1/2 bDKO and control samples, indicating that the modulatory effects of the N4-1/2 bDKO on Kir4.1 and Cx43 levels arise posttranscriptionally (Fig. 3, N and O).

## N4-2 plays a dominant role in the regulation of Kir4.1 and Cx43

We found that the levels of Kir4.1 and Cx43 in brain-specific N4-1 single KOs (*N4-1*[f/f];*EMX*-Cre; N4-1 bKO) were unchanged as compared with littermate controls (*N4-1*[f/f]; N4-1 CTL), while brain-specific N4-2 single KOs (*N4-2*[f/f];*EMX*-Cre; N4-2 bKO) showed significant increases in Kir4.1 and Cx43 as compared with littermate controls (*N4-2*[f/f]; N4-2 CTL) using the cortical

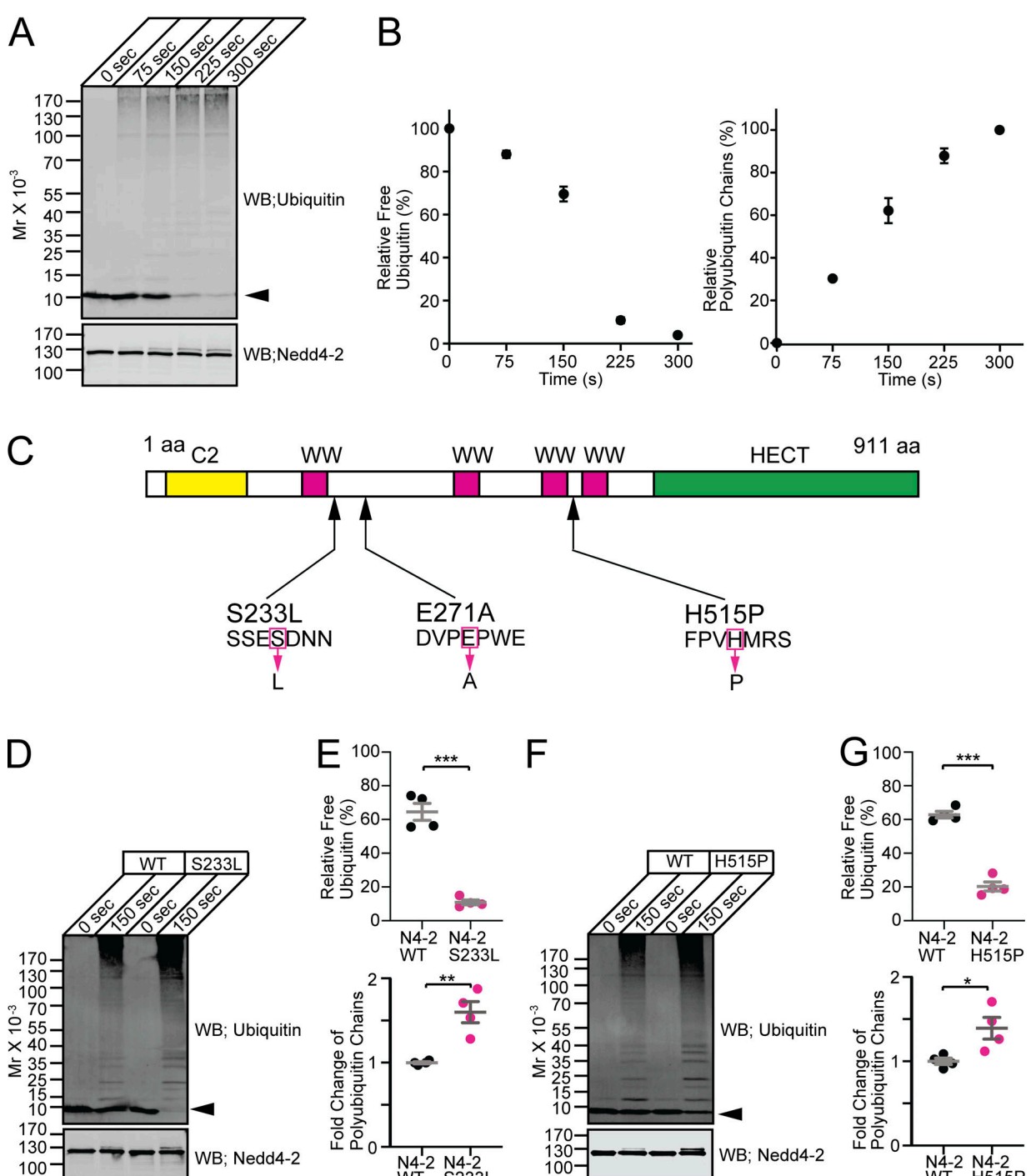

Figure 2. **Polyubiquitin chain formation activities of Nedd4-2 WT, S233L, and H515P mutants. (A)** Time course of ubiquitination using recombinant wild-type Nedd4-2 (Nedd4-2 WT). Purified Ub was incubated with ATP, E1, E2, and Nedd4-2 WT for indicated durations. Samples were subjected to Western blotting using anti-Ub (top panel) and anti-Nedd4-2 (bottom panel) antibodies. Note the time-dependent polyUb chain formation at the expense of free Ub (arrowhead). **(B)** The average time course of the depletion of free Ub (left) and formation of polyUb chains (right) in in vitro ubiquitination assay in A (*n* = 3 replicates). **(C)** Scheme of human Nedd4-2 and epileptic missense mutants. **(D–G)** S233L (D and E) and H515P (F and G) missense point mutants of Nedd4-2 cause gain-of-function of the catalytic activity. **(D and F)** Representative images of Western blotting using anti-Ub (upper panel) and anti-Nedd4-2 (lower panel) antibodies for in vitro ubiquitination assay samples using Nedd4-2 WT and mutants (S233L in D and H515P in F). Arrowheads, free Ub. **(E and G)** Quantifications of relative free Ub (top dot plots) and formation of polyUb chains relative to N4-2 WT (bottom dot plots) after 150 s incubation. Nedd4-2 WT (black dots) and Nedd4-2 mutants (magenta dots). Results are shown as mean ± SEM. The number of experiments are four for each set of experiments. ***, P < 0.001; **, 0.001 < P < 0.01; *, 0.01 < P < 0.05 (two-tailed Student's *t* test). Data distribution was assumed to be normal, but this was not formally tested. See also Table S2.

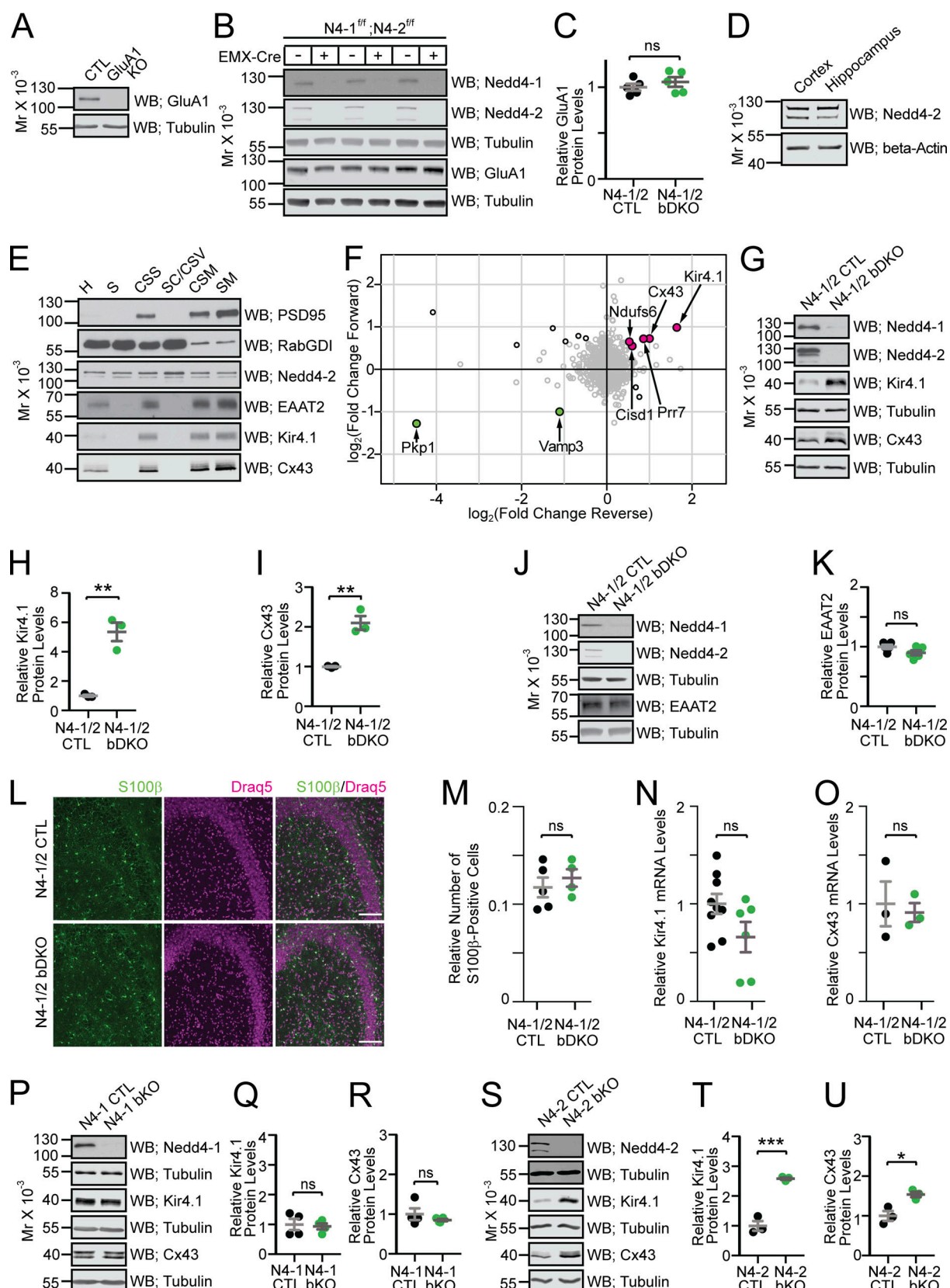

Figure 3. **Screening for proteins upregulated in Nedd4-1/2 brain-specific knockout mice. (A)** Specificity of anti-GluA1 antibody verified with cortical homogenates from CTL and GluA1 KO. **(B and C)** Quantitative Western blots with the anti-GluA1 antibody using cortical homogenates from N4-1/2 CTL and N4-1/2 bDKO mice. In C, black dots, N4-1/2 CTL; green dots, N4-1/2 bDKO. **(D)** Relative Nedd4-2 protein levels in the cortex and hippocampus in wild-type mice. **(E)** Protein profiles of PSD95, RabGDI, Nedd4-2, EAAT2, Kir4.1, and Cx43 in subcellular fractionated samples from wild-type mice. H, homogenate; S,

soluble; CSS, crude synaptosome; SC/CSV, synaptic cytoplasm/crude synaptic vesicle; CSM, crude synaptic membrane; and SM, pure synaptic membrane fractions. **(F)** Scatter plot of relative protein abundance as quantified by mass spectrometry. The $log_2$-transformed fold-change ratios between N4-1/2 bDKO and N4-1/2 CTL in the forward (y-axis) against reverse (x-axis) experiments were plotted. Black circles indicate proteins significantly changed in both experiments (significance B values <0.05). Color filling indicates proteins consistently upregulated (magenta) or downregulated (green) in the same direction in both experiments. **(G)** Representative Western blotting of cortical brain lysates from N4-1/2 CTL and N4-1/2 bDKO mice. Faint bands crossreacting with the anti-Nedd4-1 antibody in N4-1/2 bDKO samples in B and G are likely from cell types without Cre-expression (i.e., inhibitory neurons, blood cells, blood vessel cells, or microglia cells). **(H and I)** Levels of Kir4.1 (H) and Cx43 (I) in N4-1/2 CTL (black dots) and N4-1/2 bDKO (green dots). **(J and K)** Quantitative Western blotting using the anti-EAAT2 antibody showed no difference between cortical lysates from N4-1/2 CTL (black dots in K) and N4-1/2 bDKO (green dots in K). **(L)** Representative images of S100 β– and Draq5-stained CA3 regions of hippocampi from the control and N4-1/2 bDKO mice. Scale bars, 50 μm. **(M)** The number of S100β-positive cells (green channel in L) normalized to the number of Draq5-positive particles (magenta channel in L) showed no difference. **(N and O)** Kir4.1 (N) and Cx43 (O) mRNA levels were not significantly different in N4-1/2 CTL (black dots) and N4-1/2 bDKO (green dots). **(P–U)** Protein levels of Kir4.1 and Cx43 in *Nedd4-1*[f/f]; *EMX*-Cre (N4-1 bKO) and *Nedd4-1*[f/f] (N4-1 CTL) (P–R), and N4-2 bKO and N4-2 CTL (S–U) cortical lysates. Results are shown as mean ± SEM. Numbers of mice (n); (C), n = 5 for each genotype; (H and I), n = 3 for each genotype; (K), n = 5 for each genotype; (M), n = 5 for N4-1/2 CTL, n = 4 for N4-1/2 bDKO; (N), n = 9 for N4-1/2 CTL, n = 6 for N4-1/2 bDKO; (O), n = 3 for each genotype; (Q and R), n = 4 for each genotype; (T and U), n = 3 for each genotype. ***, P < 0.001; **, 0.001 < P < 0.01; *, 0.01 < P < 0.05; ns, 0.05 < P. Two-tailed Student's *t* test. Data distribution was assumed to be normal, but this was not formally tested. See also Tables S1 and S2.

lysates (Fig. 3, P–U). These results indicate that N4-2 is the dominant E3 for Kir4.1 and Cx43.

### Ubiquitination of Kir4.1 and Cx43 by N4-1 and N4-2

To test if N4-1 and N4-2 are able to ubiquitinate Kir4.1 and Cx43, we employed recombinant cell-based ubiquitination assays. Ubiquitination of HA-tagged Kir4.1 (Kir4.1-HA) or Cx43 (Cx43-HA) by EGFP-tagged Nedd4 E3s (EGFP-N4-1 and EGFP-N4-2) was readily detected by anti-Ub antibody labeling (third panels in Fig. 4, A and B). Interestingly, Ub signals from Kir4.1-HA and Cx43-HA samples were more prominent in EGFP-N4-2-overexpressing cells than in EGFP-N4-1-overexpressing cells, although EGFP-N4-1 was expressed more than EGFP-N4-2 (top panels in Fig. 4, A and B), indicating that N4-2 is a more efficient E3 for the ubiquitination of Kir4.1 and Cx43 than N4-1.

Although mammalian Nedd4 subfamily E3s form K63-linked polyUb chains (K63-Ub-chains) preferentially (Maspero et al., 2013), their yeast ortholog Rsp5 has been demonstrated to conjugate K48-linked polyUb chains (K48-Ub-chains) to substrates under heat stress (Fang et al., 2014). To investigate whether Kir4.1 and Cx43 are conjugated with K63- or K48-Ub-chains, cell-based ubiquitination samples were immunoblotted with antibodies specific for K48- and K63-Ub-chains (bottom two panels in Fig. 4, A and B). Signals from Kir4.1-HA detected by both antibodies were pronounced in an EGFP-N4-2-dependent manner, indicating that N4-2 conjugates K48- and K63-Ub-chains to Kir4.1. Cx43-HA immunoprecipitated from HEK293FT cells showed a robust signal from the anti-K48-Ub-chain antibody in the absence of recombinant N4-1 or N4-2 (the third lane in the fourth panel in Fig. 4 B). This is probably caused by the clearance of excess Cx43-HA by the endoplasmic reticulum–associated degradation (ERAD) system, where K48-Ub-chains play a crucial role. The signal from K48-Ub-chains was diminished upon overexpression of N4-1 or N4-2 (the first and second lanes in the fourth panel in Fig. 4 B), whereas K63-Ub-chains conjugated with Cx43-HA were increased (the bottom panel in Fig. 4 B). Western blotting in the bottom panel of Fig. 4 B highlights that N4-2 is the dominant E3 for Cx43, while N4-1 has a weaker ligase activity toward this substrate. We tested the impacts of missense mutations found in familial photosensitive epilepsy patients on N4-2-dependent ubiquitination of Kir4.1-HA

and Cx43-HA (Fig. 4 C). Interestingly, all missense mutations (i.e., S233L, E271A, and H536P, which correspond to human H515P mutation) enhanced ubiquitination of both substrates (bottom panels in Fig. 4 C). Accordingly, ubiquitination of Kir4.1 and Cx43 was clearly reduced in primary astrocytes derived from N4-1/2 bDKO as compared with N4-1/2 CTLs, indicating that endogenous N4-1 and N4-2 play crucial roles in the ubiquitination of Kir4.1 and Cx43 in astrocytes (Fig. 4, D and E). Small fractions of Kir4.1-HA and Cx43-HA remained ubiquitinated in N4-1/2 DKO cells (third lanes in Fig. 4, D and E) probably by other types of E3s involved in ERAD or lysosomal degradation (e.g., Hrd-1 and c-Cbl).

Using recombinant full-length N4-1, N4-2, and fragments of N4-2 (Fig. 4 F), we mapped relevant substrate binding regions. Fig. 4 G shows that approximately twofold more Kir4.1-HA and Cx43-HA bound to purified GST-tagged N4-2 than to GST-N4-1. This result together with data in Fig. 3, P–U; and Fig. 4, A and B supports the notion that N4-2 is the prominent ligase regulating the ubiquitination and levels of Kir4.1 and Cx43, while N4-1 plays a limited role. Mapping of the substrate-binding region of N4-2 demonstrated that the third and fourth WW domains of N4-2 are sufficient to bind to Kir4.1 and Cx43 (Fig. 4 G).

### Downregulation of Kir4.1 and Cx43 in astrocytes depends on N4-2 ligase activity

Next, primary cortical astrocyte cultures were prepared from N4-1/2 bDKOs and N4-1/2 CTLs, and recombinant wild-type N4-2 (N4-2 WT) or a catalytically inactive point mutant of N4-2 (N4-2 C/S) protein was expressed to study levels of Kir4.1 and Cx43 (Fig. 4 H). The reintroduction of recombinant N4-2 WT in N4-1/2 bDKO astrocytes readily restored Kir4.1 and Cx43 protein abundance to control levels, while expression of N4-2 C/S failed, demonstrating that the levels of Kir4.1 and Cx43 in astrocytes are dependent on the enzymatic activity of N4-2.

### Blockade of the endo-lysosomal pathway in astrocytes phenocopies N4-1/2 bDKO

While K48-linked polyUb chains are directly recognized by the proteasome and play crucial roles in ERAD, K63-linked chains are important for transport toward the endo-lysosomal pathway. To study whether N4-2 promotes the degradation of Kir4.1

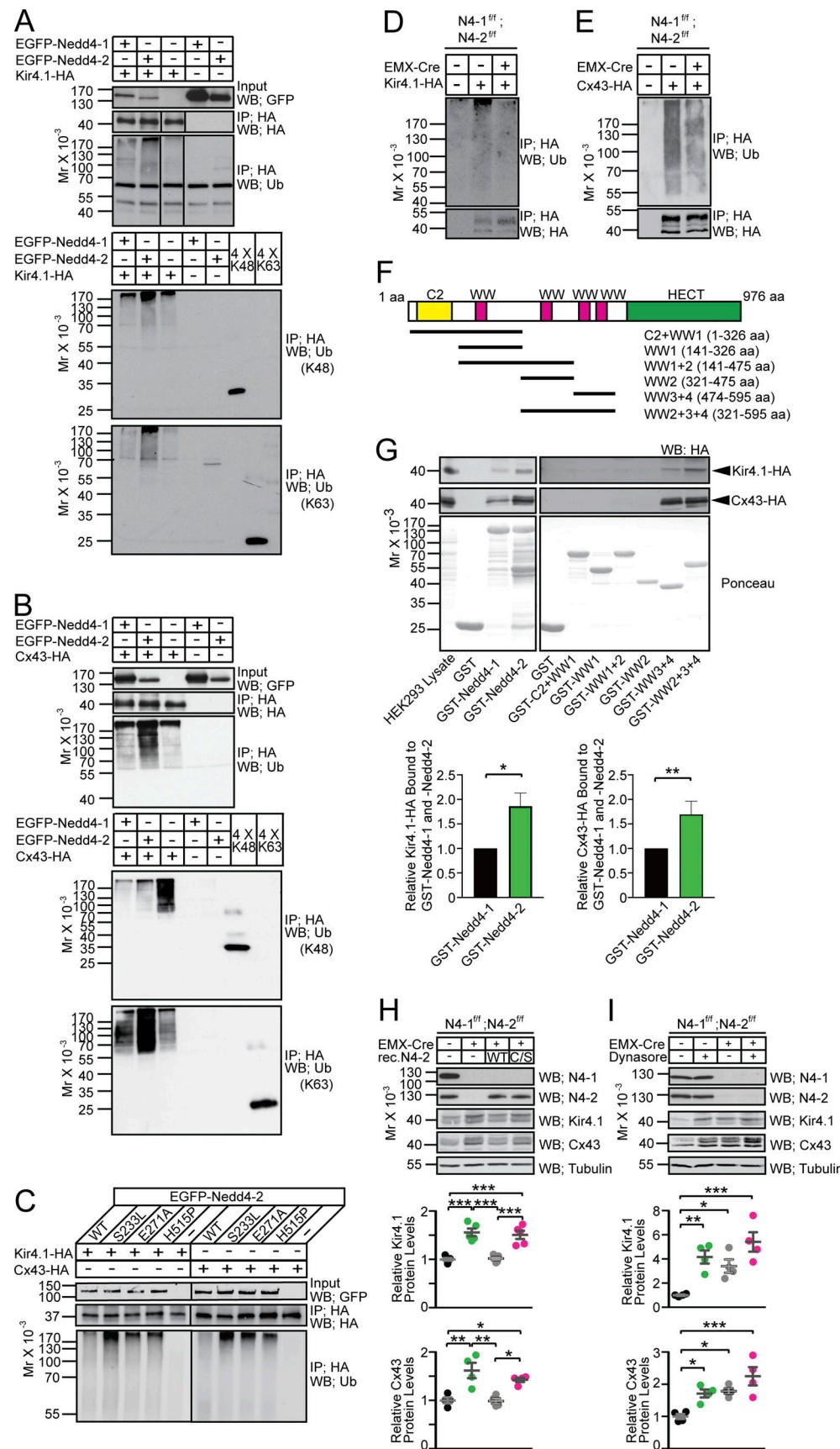

Figure 4. **Biochemical characterization of Kir4.1 and Cx43 as substrates of Nedd4-2. (A and B)** Kir4.1-HA (A) or Cx43-HA (B) was expressed in HEK293FT cells in the presence or absence of EGFP-tagged Nedd4 E3s. Levels of Nedd4 E3s were studied by an anti-GFP antibody (top panels). HA-tagged

substrates were immunoprecipitated (IP) with an anti-HA antibody and subjected to Western blotting using anti-HA (second panel) and anti-Ub (third panel) antibodies. Note that increased smear signals cross-reacting the anti-Ub antibody when EGFP-Nedd4-1 (first lane) or EGFP-Nedd4-2 (second lane) were coexpressed. Lanes in the second and third panels in A were run on the same gel but were noncontiguous. Patterns of anti-Ub Western blotting differ between (A and B) because different anti-Ub antibodies were used. The same samples were blotted with anti-K48-linked and anti-K63-linked polyUb chain antibodies (bottom two panels in A and B). Equal amounts of K48- and K63-linked tetra Ub chains were loaded in the right two lanes for SDS-PAGE together with ubiquitination assay samples. Note that signals from K48-linked and K63-linked tetra Ub chains in the bottom two blots are comparable, indicating that the anti-K48 and anti-K63 antibodies have almost the same titers. **(C)** EGFP-Nedd4-2 with one of the missense mutations found in the epileptic patients was used for the HEK293FT cell-based ubiquitination assays. H536P corresponds to human H515P. **(D and E)** N4-1/2 CTL (EMX-Cre –) and N4-1/2 bDKO (EMX-Cre +) astrocytes were infected with lentivirus expressing Kir4.1-HA (D) or Cx43-HA (E). Immunoprecipitated HA-tagged proteins were analyzed by Western blotting with anti-Ub (upper panel) and anti-HA (lower panel) antibodies. Images are representative of at least two independent experiments. **(F)** Domain structure of Nedd4-2 (accession no. NM_001114386). The amino acid sequences covered by truncated mutants of Nedd4-2 are indicated. **(G)** Affinity purification experiment using purified GST-tagged Nedd4 E3s with Kir4.1-HA (top panels) and Cx43-HA (middle panels) expressed in and extracted from HEK293FT cells. Immobilized GST-tagged proteins are stained with Ponceau (bottom panels). More Kir4.1-HA and Cx43-HA bound to GST-Nedd4-2 than to GST-Nedd4-1 (third and fourth lanes in the left top and the left middle Western blotting panels; bottom bar diagrams). Images are representative of at least two independent experiments. **(H)** Rescue of Kir4.1 (third Western blotting panel) and Cx43 (fourth Western blotting panel) levels in N4-1/2 bDKO astrocytes (EMX-Cre +) over the control (EMX-Cre –) by re-expressing recombinant wild-type Nedd4-2 (rec.N4-2 WT) but not by the inactive mutant of Nedd4-2 (rec.N4-2 C/S). In dot plots, black dots, control astrocytes; green dots, N4-1/2 bDKO astrocytes; gray dots, N4-1/2 bDKO astrocytes expressing recombinant rec.N4-2 WT; magenta dots, N4-1/2 bDKO astrocytes expressing rec.N4-2 C/S. **(I)** Upregulation of Kir4.1 (third Western blotting panel) and Cx43 (fourth Western blotting panel) by blocking endocytosis using dynasore in cultured N4-1/2 CTL astrocytes (EMX-Cre –) but not in N4-1/2 bDKO astrocytes (EMX-Cre +). In dot plots, black dots, control astrocytes treated with vehicle; green dots, control astrocytes treated with 100 µM dynasore; gray dots, N4-1/2 bDKO astrocytes treated with vehicle; magenta dots, N4-1/2 bDKO astrocytes treated with 100 µM dynasore. Results are shown as mean ± SEM. The bar diagrams in G, $n = 4$ for Kir4.1-HA binding assay and $n = 5$ for Cx43-HA binding assay; the bot plot for Kir4.1 in H, $n = 5$ for each assay point; other plots, $n = 4$ for each assay point. ***, $P < 0.001$; **, $0.001 < P < 0.01$; *, $0.01 < P < 0.05$; no asterisk, $0.05 < P$ (one-way ANOVA with Tukey's post-hoc test). Data distribution was assumed to be normal, but this was not formally tested. See also Tables S2.

and Cx43 via the endo-lysosomal pathway, clathrin-mediated endocytosis and the ensuing lysosomal degradation were blocked with dynasore, a blocker of dynamin (Macia et al., 2006), in primary astrocyte cultures (Fig. 4 I). Dynasore treatment increased the levels of Kir4.1 and Cx43 in N4-1/2 CTL astrocytes to the levels seen in N4-1/2 bDKO astrocytes, and the effects of dynasore were more pronounced in N4-1/2 CTL astrocytes than in N4-1/2 bDKO astrocytes. This indicates that dynasore phenocopies the genetic ablation of N4-1 and N4-2, and that endogenous N4-1 and N4-2 are required for the endo-lysosomal degradation of Kir4.1 and Cx43.

### Increased whole-cell currents and reduced membrane resistance in N4-2 KO astrocytes

In view of the increased steady-state levels of Kir4.1 in N4-2 bKO mice, we measured whole-cell currents and membrane resistance of cortical astrocytes embedded in acute cortical brain slices. Astrocytes stained with sulforhodamine 101 (Nimmerjahn et al., 2004) were whole-cell voltage clamped and typical whole-cell passive currents were recorded with 10 mV voltage steps from –120 to +40 mV (Fig. 5). The whole-cell inward currents of astrocytic plasma membranes were increased (Fig. 5, A, B, D, and E), and their membrane resistance was reduced (Fig. 5 F) in the absence of N4-2. These electrophysiological defects were efficiently restored by the application of Kir4.1 inhibitor fluoxetine (Fig. 5, C–F and Fig. S2, A–D), depicting that the augmented Kir4.1 level in N4-2 bKO astrocytes results in an increase in membrane ion permeability.

### Increased astrocyte coupling upon loss of N4-1 and N4-2

Next, astrocyte couplings of primary cultured N4-1/2 bDKO astrocytes, but not N4-2 bKO cells, were studied, given the potential E3 activity of N4-1 toward Cx43 (the bottom panel in Fig. 4 B). We prepared cortical but not hippocampal astrocytes,

assuming that cortical astrocytes should behave in a similar way as hippocampal astrocytes given the similar expression profiles of Cx43 and N4-2 between the two brain regions (Allen Brain Atlas; https://mouse.brain-map.org/). Primary cortical astrocyte cultures were loaded with the membrane-permeable acetomethoxy (AM) derivative of calcein (Abbaci et al., 2007). The AM ester group is cleaved off by intracellular esterases, resulting in a membrane-impermeable form of the fluorophore, calcein. Calcein is a small compound with a molecular weight of ~600 D and can diffuse through gap junctions. To estimate intercellular coupling, calcein in single cells was photobleached, and fluorescence recovery after photobleaching (FRAP) was recorded in bleached cells by time-lapse confocal microscopy. Relative FRAP in N4-1/2 bDKO astrocytes was significantly faster than that in N4-1/2 CTL astrocytes (Fig. 6), indicating that the elevation of Cx43 leads to increased syncytial connectivity in the astrocyte networks lacking N4-1 and N4-2.

### Reduced γ-oscillatory activity upon loss of N4-2 from astrocytes

To test whether altered γ-oscillatory activity shown in Fig. 1 is the direct consequence of N4-2 loss in astrocytes, we crossed N4-2 floxed mice with tamoxifen-inducible astrocyte-specific Cre driver mice (Aldh1L1-CreERT2; Winchenbach et al., 2016). Using our tamoxifen induction protocol, the specificity and efficiency of Cre recombination were confirmed by studying the expression profile of a Cre-dependent reporter protein tdTomato (tdTom) in Aldh1L1-CreERT2;Rosa26-tdTom mice (Fig. 7, A–C; Madisen et al., 2010). We confirmed that there were no tdTom-expressing cells stained with neuronal marker NeuN (650 cells in four mice) or inhibitory neuronal marker Parvalbumin (27 cells in 4 mice). The astrocyte-specific N4-2 KO mouse (N4-2$^{f/f}$; Aldh1L1-CreERT2, N4-2 AstKO) showed a significant reduction of

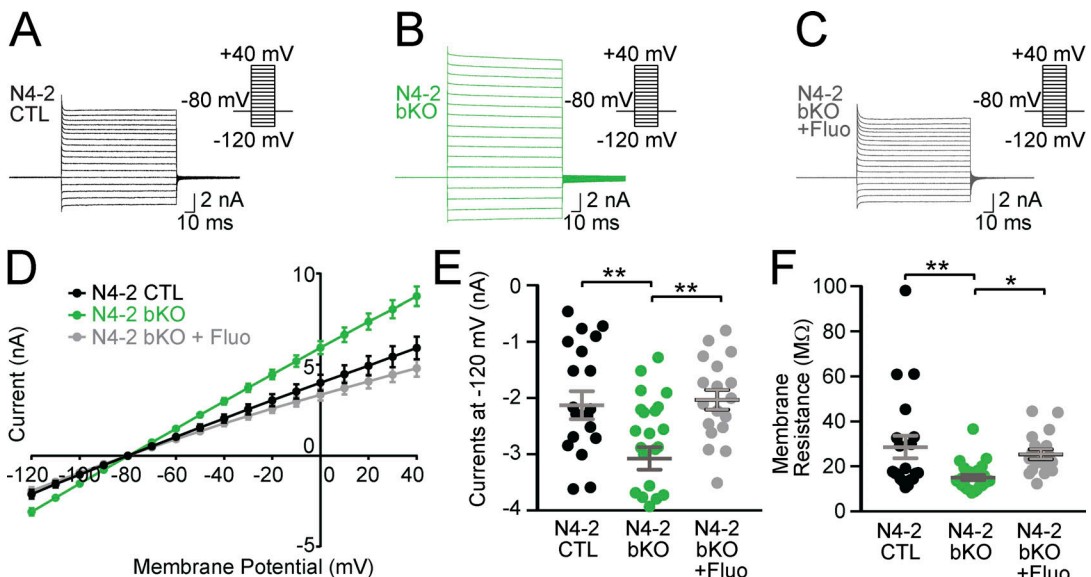

Figure 5. **Kir4.1-dependent increase in membrane conductance in N4-2 bKO astrocytes. (A–C)** Example current traces in N4-2 CTL astrocytes (A), N4-2 bKO astrocytes (B), and N4-2 bKO astrocytes treated with fluoxetine (Fluo) (C). Cells were voltage clamped at −80 and +10 mV voltage steps were applied from −120 to +40 mV during recording currents. **(D–F)** Quantifications of rescues of defects in N4-2 bKO by fluoxetine. **(D)** Voltage–current plots from N4-2 CTL (black trace), N4-2 bKO (green trace), and N4-2 bKO treated with fluoxetine (gray trace). **(E and F)** The current (E) and membrane resistance (F) at the membrane potential of −120 mV in N4-2 bKO (green dots) astrocytes were decreased as compared with N4-2 CTL astrocytes (black dots). Decreased currents (E) and membrane resistance (F) were restored to control levels by the application of Fluo in N4-2 bKO astrocytes (gray dots). Results are shown as mean ± SEM. Numbers of recorded cells (n) in D–F; n = 20 for N4-2 CTL, n = 26 for N4-2 bKO, n = 18 for N4-2 bKO + Fluo. **, 0.001 < P < 0.01; *, 0.01 < P < 0.05; no asterisk, 0.05 < P. (One-way ANOVA with Tukey's post-hoc test for E and F). Data distribution was assumed to be normal, but this was not formally tested. See also Fig. S2 and Table S2.

N4-2 protein expression and increases in Kir4.1 and Cx43 levels (Fig. 7, D–G), as well as Kir4.1-dependent increase in currents of the astrocyte plasma membrane (Fig. 7, H–M). The number of parvalbumin (PV)-positive interneurons, which are also critically involved in hippocampal CA3 γ-oscillations (Cardin et al., 2009; Gloveli et al., 2005), was found to be unchanged in the CA3 region of N4-2 AstKOs (Fig. 7, N–P). As observed in N4-2 bKO (Fig. 1, E–H), the average power of γ-oscillatory activity was significantly reduced in N4-2 AstKO as compared with N4-2 CTL (Fig. 8). The power of γ-oscillations was also reduced in the hippocampal CA3 region in anesthetized N4-2 AstKO mice in vivo (Fig. 9). These findings show that the loss of N4-2 specifically in astrocytes leads to decreased γ-oscillations in the hippocampus without influencing the number of PV-positive interneurons.

To investigate direct involvements of increases in Cx43 and Kir4.1 levels in the reduction of γ-oscillatory activity in N4-2 AstKOs, γ-oscillatory activity was recorded from CA3 hippocampal regions of acute brain slices in the presence of fluoxetine or GAP26. GAP26 is a peptide that corresponds to a part of the extracellular loop of Cx43 and thereby blocks Cx43 hemichannels and gap junctions (Chaytor et al., 1997). Both, fluoxetine (Fig. 8 C and Fig. S2, E–H) and GAP26 (Fig. 8 H) partially rescued the effect of N4-2 loss on γ-oscillations, without influencing the oscillation frequency (Fig. 8, D, E, I, and J). We concluded that the N4-2–mediated downregulation of Kir4.1 and Cx43 in astrocytes is required for the maintenance of γ-band oscillatory activity in the CA3 region of the hippocampus in parallel or in tandem (Fig. 10).

## Discussion

The present study (1) describes a novel cell biological mechanism by which N4-2 ubiquitinates two key astrocytic channel proteins, Kir4.1 and Cx43, thereby controlling their surface expression to regulate astrocyte membrane K⁺ conductance and gap junction coupling (Figs. 3, 4, 5, and 6), and (2) shows that this N4-2–dependent control of Kir4.1-mediated and Cx43-mediated astrocyte functions is a key determinant of synchronous nerve cell activity in the hippocampus (Figs. 1, 7, 8, and 9). Beyond the biological significance of our findings described above, our study provides a possible mechanistic explanation for the etiopathology of photosensitive epilepsy in patients with point mutations in N4-2 (Fig. 2, Fig. 4 C, and Fig. 10).

### Identification of N4-2 E3 ligase substrates in the brain

Our proteomic screening approach identified the two astrocytic proteins Kir4.1 and Cx43 as the most strongly upregulated proteins in the N4-1/2 bDKO brain (Table S1 and Fig. 3 F). Subsequent biochemical (Fig. 3, G–I) and qRT-PCR analyses (Fig. 3, N and O) verified that Kir4.1 and Cx43 are bona fide substrates of N4-2. Although our biochemical validation of proteome results in Fig. 3, G–I and P–U were performed using cortical lysate, it is likely that Kir4.1 and Cx43 are both substrates in hippocampal astrocytes given a wide-spread expression of these substrates and N4-2 in the brain tissue. Indeed, the ratios of hippocampal and cortical expressions of Nedd4-2, Kir4.1, and Cx43 are similar (Allen Brain Atlas; https://mouse.brain-map.org/). These data identify a new level of regulation of the major astrocytic K⁺

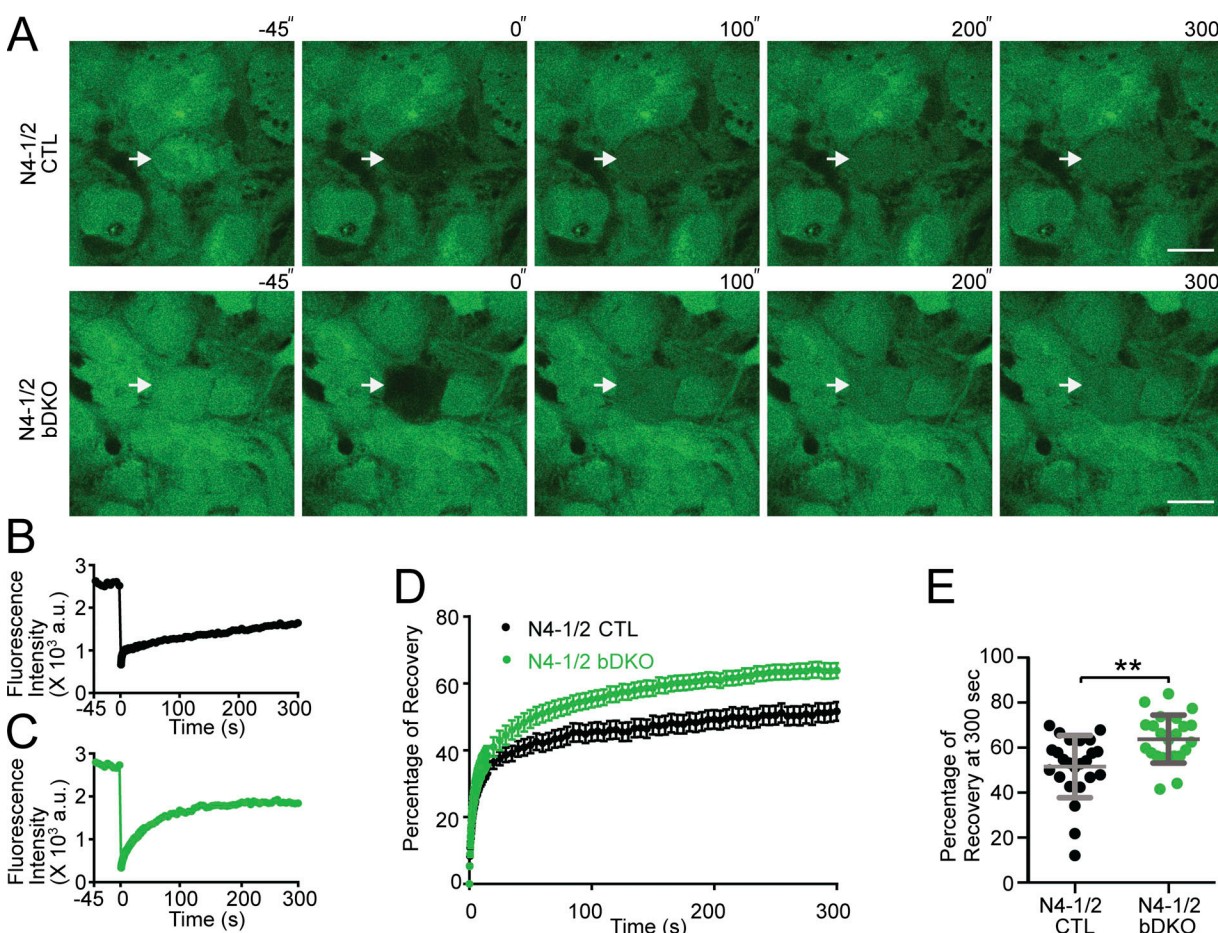

Figure 6. **Augmented astrocyte coupling in the absence of Nedd4 E3s. (A)** Representative images of FRAP live imaging experiments in N4-1/2 CTL (top panels) and N4-1/2 bDKO astrocytes (bottom panels). Primary astrocytes prepared from N4-1/2 CTL and N4-1/2 bDKO mice were loaded with calcein-AM. Calcein in the astrocytes indicated with arrows was bleached, and FRAP was recorded. Scale bars, 20 μm. **(B and C)** Time courses of FRAP at bleached cells in A for N4-1/2 CTL (B) and N4-1/2 bDKO (C) astrocytes. **(D)** Average FRAP time courses of N4-1/2 CTL (black trace) and N4-1/2 bDKO (green trace) astrocytes. **(E)** Average fluorescence recoveries at 300 s after bleaching. N4-1/2 bDKO showed a significant increase in the recovery of calcein fluorescence over N4-1/2 CTL. Results are shown as mean ± SEM. Numbers of imaged cells (*n*); *n* = 24 for N4-1/2 CTL, *n* = 23 for N4-1/2 bDKO. **, $0.001 < P < 0.01$ (two-tailed Student's *t* test). Normality of the distribution of was confirmed with the Kolmogorov–Smirnov test. See also Table S2.

channel Kir4.1 that operates posttranslationally, i.e., downstream of previously characterized transcriptional regulation (Farmer et al., 2016; Kelley et al., 2018), to control astrocyte function. Disruption of such Kir4.1 regulations could be the basis of several neurological disorders, including amyotrophic lateral sclerosis (Kelley et al., 2018) and depression (Cui et al., 2018). A recent report shows that N4-2-dependent ubiquitination of Kir4.1 is relevant for the maintenance of K+ conductance in the kidney although proteome change in kidney-specific N4-2 KO is not shown (Wang et al., 2018). The fold change of Kir4.1 in kidney-specific N4-2 KO (1.9-fold increase in Wang et al. [2018]) is clearly smaller than what we found in N4-2 AstKO (>3-fold increase in Fig. 7 F), indicating the cell-type-specific regulation of Kir4.1 by N4-2 in astrocytes.

## Consequences of N4-2 loss on astrocyte function

Most E3 ligases are characterized by a rather broad substrate spectrum, and many proteins can be ubiquitinated by multiple E3s. Nevertheless, the major consequences of loss-of-function of a certain E3 in a given cell type are often due to aberrant

ubiquitination of only a few substrates (Hengstermann et al., 2005; Tokunaga et al., 2009), indicating a substantial cell-type specificity and selectivity of E3-substrate interactions. The epithelial Na+ channel ENaC, one of the most prominent and best-established substrates of mammalian Nedd4-family E3s (Staub et al., 1996), is ubiquitinated by N4-2 (Kamynina et al., 2001). ENaC ubiquitination and its subsequent downregulation are of particular importance for normal membrane Na+ conductance in kidney epithelial cells. Liddle syndrome patients, who suffer from hypertension, have frame-shift or point mutations in the *ENaC* gene that disrupt ENaC-binding to N4-2 (Shimkets et al., 1994; Tamura et al., 1996), leading to a loss of ubiquitination and consequent upregulation of ENaC (Firsov et al., 1996). In rodents, loss of N4-2 is accompanied by salt-induced hypertension that mimics the Liddle syndrome phenotype in terms of altered ENaC-ubiquitination, ion balance, and blood pressure, confirming the critical and rather specific role of N4-2 in the regulation of ENaC levels and function in the kidney (Minegishi et al., 2016; Ronzaud et al., 2013).

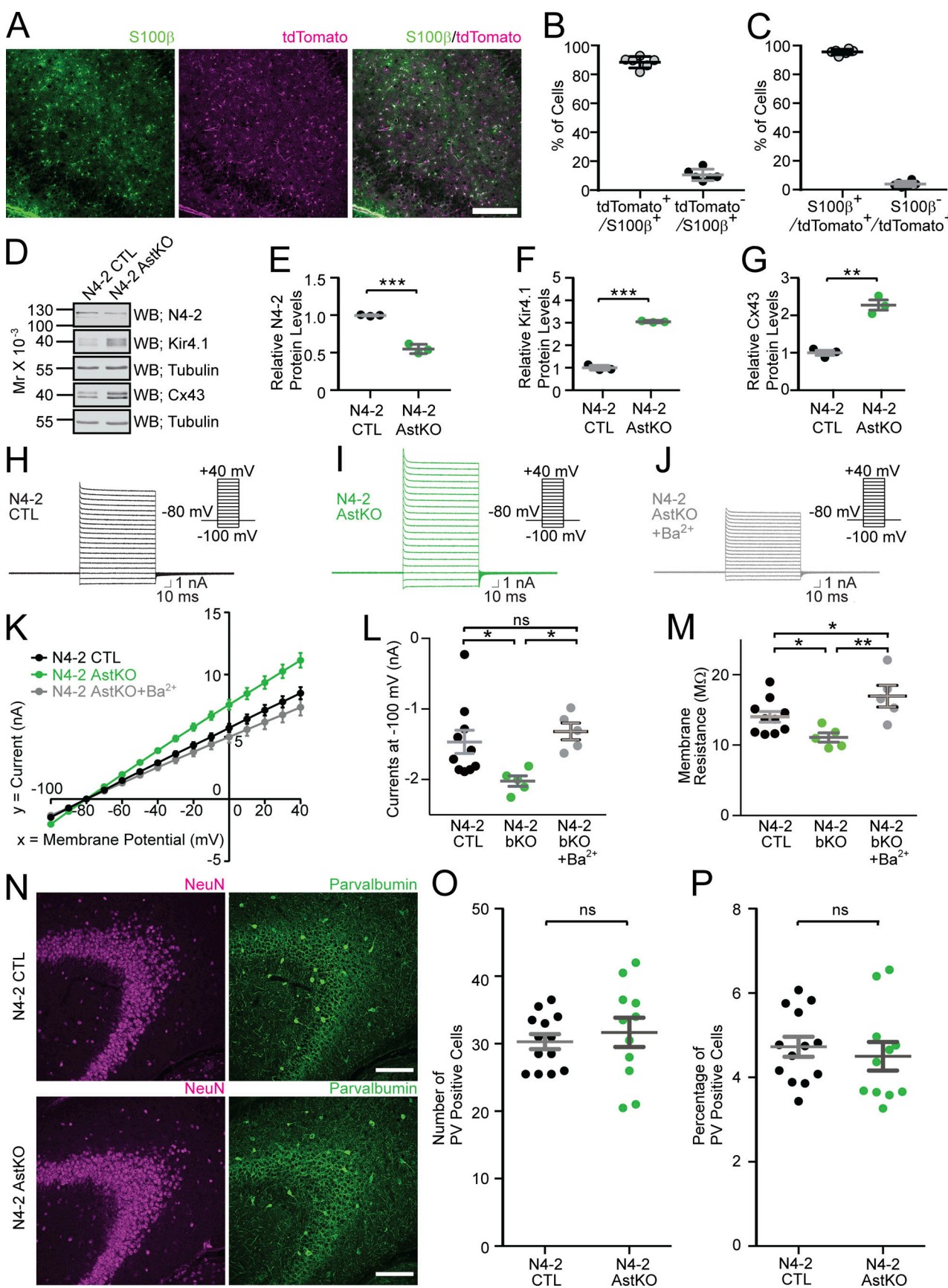

Figure 7. **Characterization of astrocyte-specific N4-2 conditional KO. (A)** *Aldh1l1*-CreERT2 mouse crossed with tdTomato-expressing Cre indicator mouse (*Aldh1l1*-CreERT2;*ROSA26*-tdTom) was injected with tamoxifen to induce Cre recombination with the same protocol used for *Nedd4-2^f/f^;Aldh1l1*-CreERT2 (N4-2

AstKO) mice. The hippocampal CA3 region was immunostained for an astrocyte marker S100β. Scale bar, 200 µm. **(B)** The efficiency of Cre recombination in astrocytes. 88.5 ± 1.5 (SEM) % of S100β-positive cells are also positive for tdTomato. **(C)** The specificity of Cre recombination. 95.1 ± 0.77 (SEM) % of tdTomato-expressing Cre-recombined cells are positive for S100β. The total number of cells counted in B and C is 898 altogether. Six images taken from three mice were analyzed. **(D)** Representative Western blotting results using cortical lysates from N4-2 CTL and N4-2 AstKO with antibodies to N4-2, Kir4.1, Tubulin, and Cx43. **(E–G)** Quantification of relative N4-2, Kir4.1, and Cx43 protein levels in cortical lysates. **(H–M)** An increase in plasma membrane currents in N4-2 AstKO was restored by blocking Kir4.1 with barium chloride (Ba$^{2+}$). Experiments were performed in a way similar to those in Fig. 5. All mutants were crossed with *ROSA26*-tdTom. **(H–J)** Example trances of N4-2 CTL crossed with *ROSA26*-tdTom (H), N4-2 AstKO crossed with *ROSA26*-tdTom (I), and N4-2 AstKO crossed with *ROSA26*-tdTom and treated with Ba$^{2+}$ (J). **(K–M)** Voltage-current plots (K), currents at −100 mV (L), and membrane resistance (M) from three conditions. **(N–P)** Unchanged number of parvalbumin-positive cells in N4-2 AstKO. **(N)** Representative images of N4-2 CTL (top panels) and N4-2 AstKO (bottom panels) hippocampal sections stained with anti-Neuronal Nuclei (NeuN) and anti-Parvalbumin (PV) antibodies. Scale bars, 300 µm. **(O and P)** Absolute numbers of PV-positive cells in each entire imaged field (O) and the percentages of PV-positive cells with respect to total NeuN-positive cells (P). The number of mice for (E–G) is three for each genotype. Numbers of recorded cells (*n*) in K–M; *n* = 10 for N4-2 CTL, *n* = 5 for N4-2 AstKO, *n* = 5 for N4-2 AstKO + Ba$^{2+}$. Numbers of brain sections in O and P (*n*); *n* = 13 for N4-2 CTL, *n* = 11 for N4-2 AstKO. ***, $P < 0.001$; **, $0.001 < P < 0.01$; *, $0.01 < P < 0.05$; ns, $0.05 < P$ (two-tailed Student's *t* test for E–G, O, and P; one-way ANOVA with Newman–Keuls test for L and M). The normality of the distribution was confirmed with the Kolmogorov–Smirnov or Shapiro–Wilk test. See also Table S2.

The present study demonstrates that a dominant function of N4-2 in the adult brain is to ubiquitinate and downregulate Kir4.1 and Cx43 in astrocytes to control neuronal network activity. Loss of N4-2 leads to elevation of Kir4.1 and Cx43 protein levels (Fig. 3, S–U and Fig. 7, D–G), causing increases in the membrane ion permeability and in gap junction connectivity of astrocytes (Figs. 5 and 6 and Fig. 7, H–M). Astrocytic K$^+$ uptake is mediated by three major processes involving Kir4.1, Na$^+$-K$^+$-Cl$^-$- cotransporters, and Na$^+$/K$^+$-ATPase. Although substantial amounts of K$^+$ are released by neurons during their repolarization

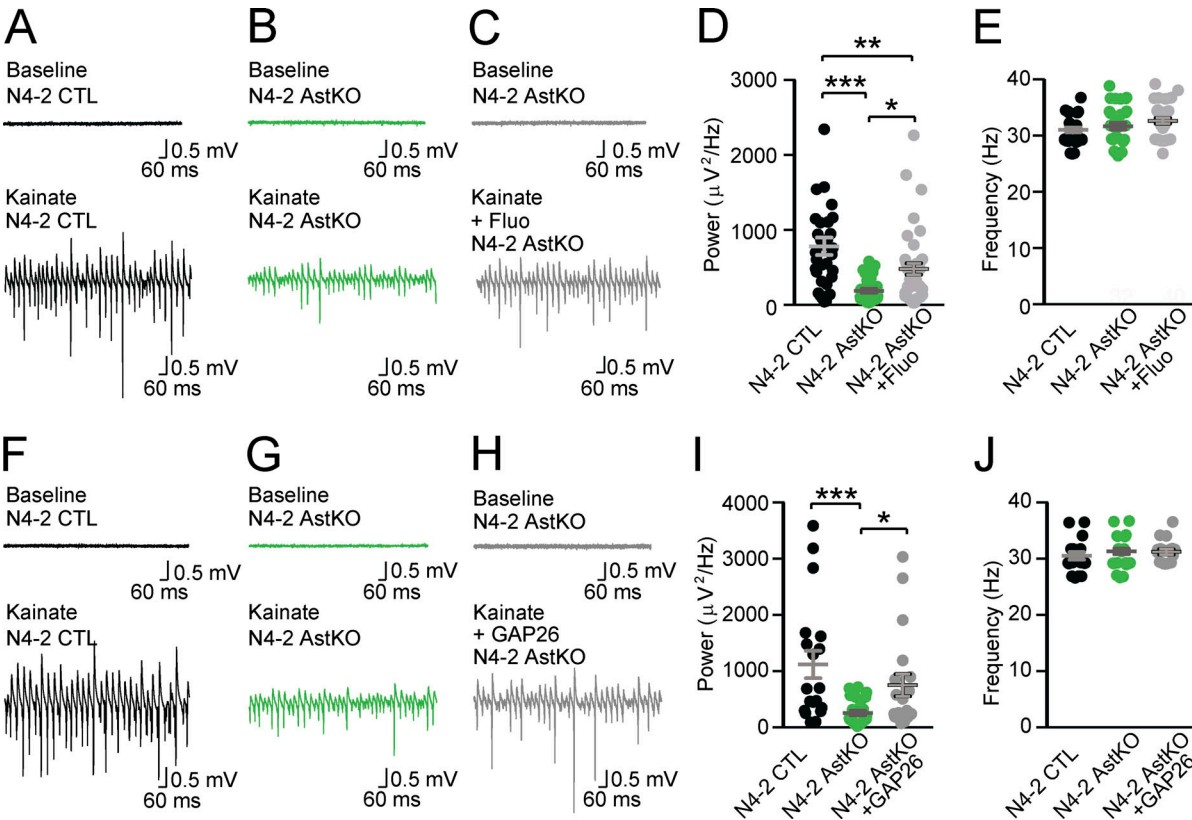

**Figure 8.  Rescue of reduced γ-oscillation in N4-2 AstKO by pharmacological inhibition of Kir4.1 or gap junction. (A and B)** Representative recordings in the CA3 region of acute hippocampal slices from N4-2 CTL (A) and N4-2 AstKO (B) mice before (baseline) and after (Kainate) induction of γ-oscillations with 100 nM kainate application. **(C)** Representative recordings in the CA3 region of acute hippocampal slices from N4-2 AstKO mice before (baseline) and after (Kainate + Fluo) induction of γ-oscillations with 100 nM kainate application in the presence of fluoxetine. **(D and E)** Average powers (D) and frequencies (E) of γ-oscillations in N4-2 CTL (black dots) and N4-2 AstKO without (green dots) and with (gray dots) fluoxetine. **(F–H)** Representative recordings from the same set of experiments as (A–C), except for the usage of GAP26 as a pharmacological blocker in H. **(I and J)** Average powers (I) and frequencies (J) of γ-oscillations in N4-2 CTL (black dots) and N4-2 AstKO hippocampal slices without (green dots) and with (gray dots) GAP26. Results are shown as mean ± SEM. Numbers of recorded slices (*n*); (D and E), *n* = 33 for N4-2 CTL, *n* = 32 for N4-2 AstKO, *n* = 40 for N4-2 AstKO + Fluo; (I and J), *n* = 19 for N4-2 CTL, *n* = 27 for N4-2 AstKO, *n* = 19 for N4-2 AstKO + GAP26. ***, $P < 0.001$; **, $0.001 < P < 0.01$; *, $0.01 < P < 0.05$; no asterisk, $0.05 < P$ (one-way ANOVA with Newman–Keuls test for D, E, I, and J). Data distribution was assumed to be normal, but this was not formally tested. See also Fig. S2 and Table S2.

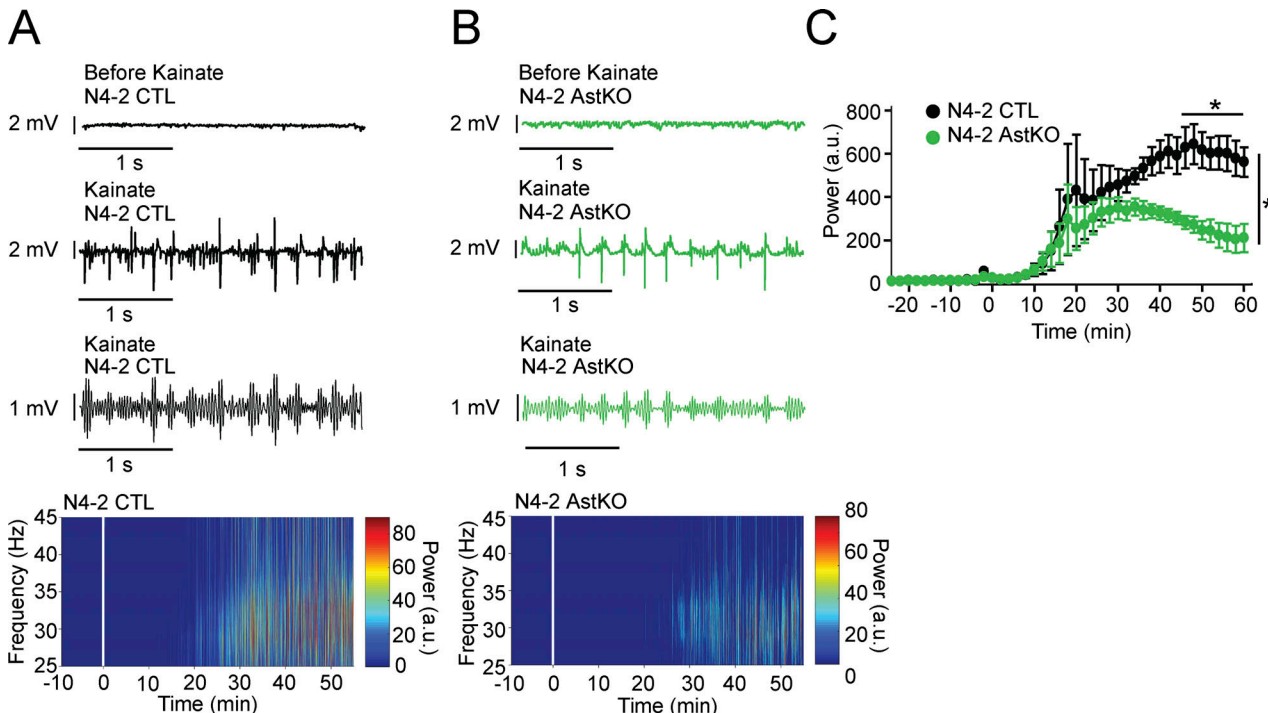

Figure 9. **Decreased γ-oscillations in N4-2 AstKO mice in vivo. (A and B)** Representative local field potentials (LFP) were recorded in the CA3 region of the hippocampus in vivo from anesthetized N4-2 CTL (A) and from N4-2 AstKO (B) mice before (Before Kainate) and after (Kainate) intraperitoneal injection of kainate. Third traces in A and B are representative slow-γ filtered LFPs. Bottom heat maps are representative normalized spectrograms. Kainate was injected at time point 0 min. **(C)** The normalized average power of slow-γ-band oscillation in N4-2 CTL (black trace) and N4-2 AstKO (green trace) mice. Slow γ-band oscillation in N4-2 AstKO showed a siginificantly reduced power in comparison to the N4-2 CTL. Results are shown as mean ± SEM. N = 4 animals for N4-2 CTL and N = 4 animals for N4-2 AstKO. *, P < 0.05 (repeated measures two-way ANOVA with LSD post-hoc test). Data distribution was assumed to be normal, but this was not formally tested. See also Table S2.

after action potential firing, $[K^+]_o$ never exceeds the approximate ceiling level of ∼12 mM, not even during prolonged neuronal network activity (Heinemann and Lux, 1977). This is thought to be at least partly due to the fact that locally elevated $[K^+]_o$ is removed via Kir4.1 channels on the plasma membrane of astrocytes (Higashi et al., 2001; Karwoski et al., 1989), followed by the dissipation of the consequently and transiently increased astrocytic $K^+$ levels within the syncytial astrocytic network via gap junctions (Wallraff et al., 2006). Our data show that N4-2-dependent ubiquitination limits the maintenance of Kir4.1 and Cx43 on the astrocytic plasma membrane and promotes their lysosomal degradation so that spatial $K^+$ buffering by astrocytes is increased in the absence of N4-2 at two distinct levels, i.e., at the astrocyte surface via increased Kir4.1 activity and at astrocytic gap junctions via increased Cx43 levels (Fig. 10).

### Non-cell-autonomous roles of N4-2 in neuronal network synchronicity

The decreased γ-oscillations seen in N4-2 AstKO are reverted to normal levels by fluoxetine (Fig. 8). Fluoxetine is relatively selective for Kir4.1, with only minor effects on other Kir channels at the concentration used in our experiments (30 µM; Ohno et al., 2007). It still could have acted as a serotonin uptake inhibitor in our experiments, causing increases in extracellular serotonin levels (Malagié et al., 1995). However, it is unlikely that increased extracellular serotonin levels restore the

phenotypic change in γ-oscillations in the N4-2 AstKO because bath-application of serotonin actually blocks kainate-induced γ-oscillations in rat hippocampal slices (Wójtowicz et al., 2009). Thus, we conclude that the increased astrocytic Kir4.1 levels are a major cause for the reduced power of γ-oscillations in N4-2 AstKOs and that the phenotype-reverting effect of fluoxetine is mediated by a blockade of the excess Kir4.1 channels. It was reported recently that oligodendrocyte-specific Kir4.1 conditional KO showed impaired $K^+$ clearance and axonal degeneration in the white matter (Larson et al., 2018; Schirmer et al., 2018). In *Aldh1L1*-CreERT2;*Rosa26*-tdTom mice, Cre recombination takes place almost exclusively in astrocytes (Fig. 7, A–C) and only rarely in other cell types, such as oligodendrocytes or neurons (Winchenbach et al., 2016). The increase in Kir4.1 levels in N4-2 AstKO is similar in magnitude to that seen in N4-2 bKO (compare Fig. 3 T and Fig. 7 F). These results indicate that N4-2 controls Kir4.1 levels predominantly in astrocytes.

Like fluoxetine, GAP26 reverts the reduced power of γ-oscillations in N4-2 AstKOs (Fig. 8). In principle, this effect of GAP26 could be due to blockade of either hemichannels or gap junctions—so far, it is not possible to block only gap junctions without affecting hemichannels by using blocking peptides or genetic approaches (e.g., knock-down or KO of *Cx43*). In our FRAP experiments, the calcein signal decayed without bleaching likely because of calcein release through hemichannels.

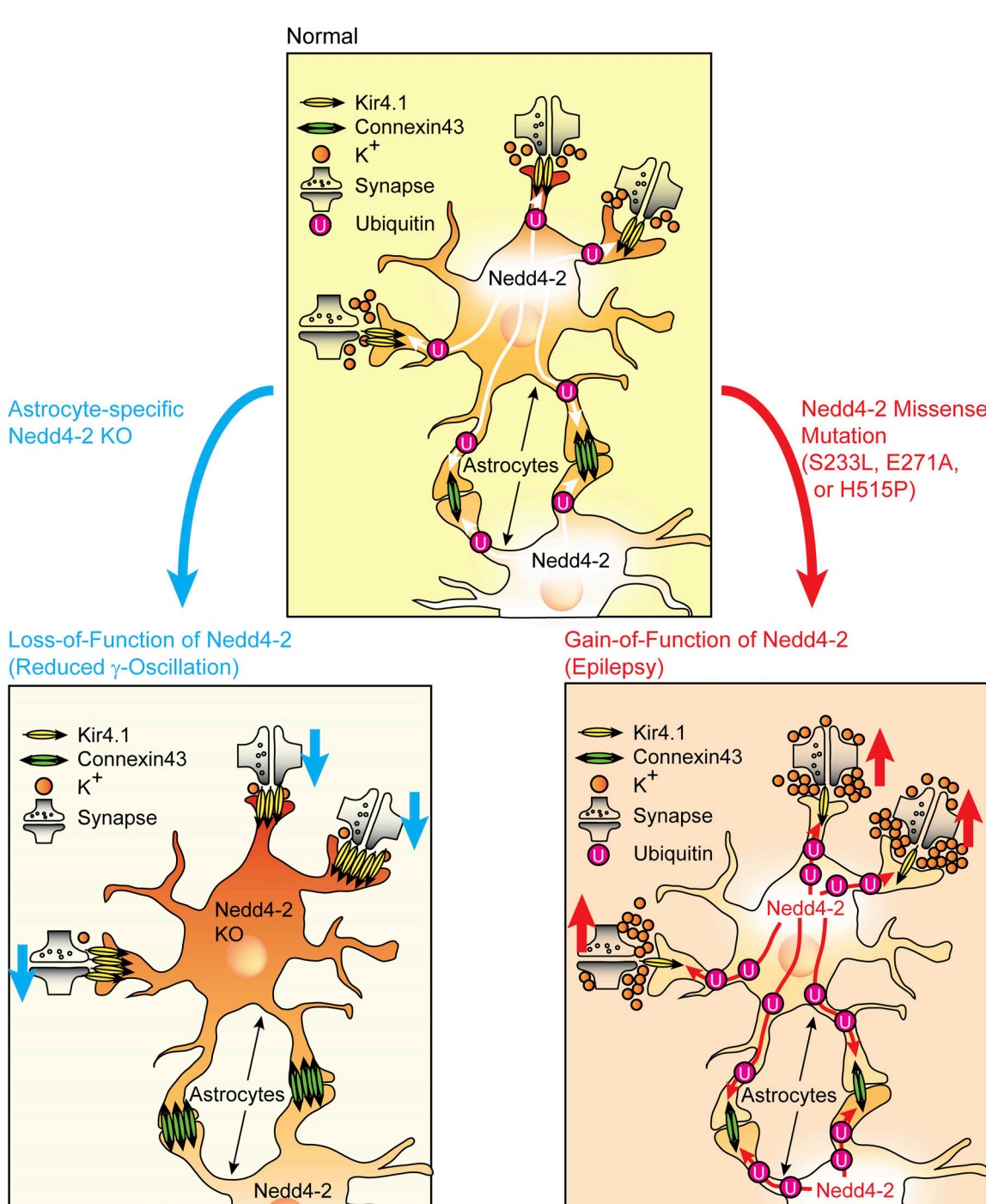

Figure 10. **Physiological and pathophysiological homeostasis of astroglial channel proteins by Nedd4-2.** Top: Kir4.1 and Cx43 are ubiquitinated by Nedd4-2 and thus degraded by lysosome in astrocytes. This process limits the uptake of extracellular potassium and the subsequent dissipation of intracellular potassium via gap junctions. Sustained extracellular potassium is critical for the maintenance of the neuronal network activity. Left bottom: In astrocyte-specific Nedd4-2 KO mouse, the neuronal network activity is depressed because of reduced extracellular potassium caused by the augmentation of Kir4.1 and Cx43 levels in astrocytes. Right bottom: Nedd4-2 missense mutation (S233L, E271A, or H515P) causes the gain-of-function of E3 ligase activity, increasing ubiquitination levels of Kir4.1 and Cx43, and thus reducing their protein expression. This is the potential cause of an enhanced neuronal network activity and epilepsy.

However, the time course of this decay (∼30 min) was longer than that of FRAP (∼5 min), indicating that gap junctions play a more prominent role than hemichannels in the astrocytic efflux of small solutes such as calcein, ions, or metabolic substrates.

In the N4-2 AstKO, the increased Cx43 levels could equally affect inter-astrocytic gap junctions composed of Cx43, gap junctions composed of Cx43 and Cx36 between astrocytes and neurons, and gap junctions composed of Cx43 and Cx47 between

astrocytes and oligodendrocytes. Through upregulated gap junctions in N4-2 AstKO, the excitability of neurons might be influenced by the cumulative effect of enhanced $K^+$ buffering (Wallraff et al., 2006), increased lactate delivery to neurons (Clasadonte et al., 2017), and increased extracellular space volume (Pannasch et al., 2011), all of which might reduce the power of $\gamma$-oscillations. Of these possible explanations, we regard $K^+$ buffering as the most crucial. Even transient elevation of extracellular $K^+$ causes oscillatory activity in the $\gamma$-band frequency without much effect on other brain oscillation frequencies (LeBeau et al., 2002), which would be compatible with our results (Figs. 8 and 9).

$\gamma$-oscillations precede epileptiform spike events (Ren et al., 2015), and visual stimuli that cause maximum power of $\gamma$-oscillations could trigger photosensitive epilepsy (Hermes et al., 2017; Perry et al., 2014). In this context, we found that missense mutations in N4-2 that were identified in patients with photosensitive epilepsy (S233L, E271A, and H515P) cause increased intrinsic E3 activity of N4-2 (Fig. 2) and/or N4-2-dependent ubiquitination of Kir4.1 and Cx43 (Fig. 4 C). This result indicates that the corresponding mutant N4-2 variants might cause reduced astrocytic expression of Kir4.1 and Cx43, reduced $K^+$ buffering, and correspondingly increased neuronal network activity, which could underlie the pathology of the patients carrying these N4-2 mutations (Bockenhauer et al., 2009; Djukic et al., 2007; Scholl et al., 2009; Wallraff et al., 2006; Fig. 10).

In conclusion, our study defines astrocytic proteostasis as an important determinant of astrocyte function and as a novel regulatory principle in neuronal network activity that may have major implications for the etiology of certain forms of epilepsy.

## Materials and methods

### Details of antibodies
Details of antibodies are described in Table S3.

### Animal experiments
All experiments using mice were performed at Max Planck Institute for Multidisciplinary Sciences in compliance with the guidelines for the welfare of experimental animals approved by the State Government of Lower Saxony (Niedersächsisches Landesamt für Verbraucherschutz und Lebensmittelsicherheit Permission 33.9-42502-04-13/1359, 33.19-42502-04-15/1954, and 33.19-42502-04-16/2173) and the Max Planck Society (comparable to National Institute of Health Guidelines). For all animal experiments, mice backcrossed with the C57BL/6N >10 times were used. *Nedd4-2*^f/f;*Aldh1l1*-CreERT2 and *Nedd4-2*^f/f male mice were injected with 50 µl of 10 mg/ml tamoxifen for five consecutive days starting at P13 to P15. At the age of 5 wk, mice were anesthetized by intraperitoneal injection of urethane (1.3–1.5 mg/kg body weight) prior to in vivo recording.

### In vitro recording of γ-oscillations
Recordings of LFPs from acute hippocampal slices and the induction of oscillations were performed as previously described (Ripamonti et al., 2017). The frequency at maximum power peak

and average power of oscillations were determined between 25 and 45 Hz. For rescue experiments performed under conditions of Kir4.1 inhibition, the baseline spectrum was recorded in artificial cerebrospinal fluid (ACSF) and $\gamma$-oscillation was induced with Kainate solution (100 nM Kainate in ACSF) in the presence of 30 µM fluoxetine, which inhibits Kir4.1 channel (Cat #PHR1394-1G; Sigma-Aldrich). Since GAP26 has a larger molecular weight than fluoxetine, it is generally required to incubate the tissue with this inhibitor prior to functional experiments to achieve efficient penetration of GAP26 into the tissue. For rescue experiments performed under conditions of Cx43 inhibition, slices were preincubated for 45–60 min in ACSF containing 50 µM the Cx43 inhibitor GAP26, the baseline spectrum was measured in the same solution, and $\gamma$-oscillation was induced by Kainate solution containing 50 µM GAP26. Each set of pharmacological rescue experiments (i.e., recordings from N4-2 CTL, N4-2 AstKO, and N4-2 AstKO treated with a blocker) was performed as a single set of recording experiments.

### Quantitative mass spectrometry
SM fractions were purified as previously reported (Mizoguchi et al., 1989). Proteins in pooled SM fractions from N4-1/2 CTL, N4-1/2 bDKO, and a 1:1 mixture of both were separated on precast gradient gels and visualized by colloidal Coomassie staining. Proteins in the gel were subjected to in-gel digestion and isobaric peptide labeling as described (Schmidt et al., 2013). The iTRAQ Reagent 4plex Kit (Cat#4352135; Sciex) was used to label the tryptic peptides derived from the different SM fractions as follows: iTRAQ114, mixture of equal volumes of all three samples to be compared; iTRAQ115, 1:1 mixture of N4-1/2 CTL and N4-1/2 bDKO; iTRAQ116, N4-1/2 CTL; and iTRAQ117, N4-1/2 bDKO (forward labeling). To control for reagent-specific labeling artifacts, a replicate experiment was performed in which iTRAQ channels 116 and 117 were switched (reverse labeling). The iTRAQ114/115 channels served as internal controls for normalization and quality control of the iTRAQ reporter ion signals. Labeled peptides were pooled and analyzed by liquid chromatography coupled to electrospray mass spectrometry using an LTQ Orbitrap Velos hybrid mass spectrometer (Thermo Fisher Scientific) operated in a data-dependent mode with higher-energy C-trap fragmentation as described (Schmidt et al., 2013). MS raw data were processed with the MaxQuant software (version 1.3.0.5) and peak lists were searched with the built-in Andromeda search engine (Cox and Mann, 2008) against UniProtKB *M. Musculus* protein database (downloaded 2013-05-14) supplemented with common contaminants and concatenated with the reverse sequences of all entries. Search parameters were set to carbamidomethylation of cysteines as fixed and oxidation of methionine and N-terminal acetylation as variable modifications. Trypsin without proline restriction was specified as a digestion enzyme and up to two missed cleavages were allowed. The precursor and the fragment ion mass tolerance were set to 7 and 20 ppm, respectively. A minimal length of six amino acids per identified peptide was required for protein identification. The false discovery rate was set to 1% at both peptide and protein levels. The command "re-quantify" was enabled and "keep low scoring versions of identified peptides"

was disabled. Statistical analysis was performed with the Perseus bioinformatics platform to calculate "Significance B" (Tyanova et al., 2016). For candidate selection, only proteins with a minimum of two identified peptides in total (at least one unique) were considered and consistent up- or down-regulation in both experiments (forward and reverse labeling) was required. Upregulated proteins with a "Significance B" value below 0.05 in both experiments were considered potential substrates.

## Quantitative Western blotting

Quantitative Western blotting was performed as previously reported (Hsia et al., 2014). Secondary antibodies labeled with fluorescence (i.e., IRDye700 or IRDye800) were used for this purpose. Signals from secondary antibodies were acquired and quantified by an Odyssey Imaging System (LI-COR). Signals were normalized to Tubulin levels. Brain lysates from $GluA1^{-/-}$ and the control mice were provided by Drs. Thorsten Bus and Rolf Sprengel (Max Planck Institute for Medical Research, Heidelberg, Germany; Zamanillo et al., 1999). All values from one set of experiments were normalized to the average value of CTL before statistical analyses. Samples were excluded from statistical analyses when bands in Western blotting were not isolated or were obviously deformed. Averages, SEMs, P values, and statistical tests are documented in Table S2. Sample numbers are documented within the legends of each figure.

## Quantitative RT-PCR

Cortices were dissected from CTL and N4-1/2 bDKO mice with tools cleaned with 70% ethanol (Sigma-Aldrich) and RNA-Zap (Thermo Fisher Scientific) and flash-frozen in liquid nitrogen. Samples were homogenized in 600–1,000 µl of Trizol (Thermo Fisher Scientific) in 2-ml Eppendorf tubes by using Ultra Turrax homogenizer (IKA Labtechnik) for 40 s and incubated at room temperature for 3 min. Samples were centrifuged at 10,000 rpm for 2 min, and supernatants were collected and mixed with an equal volume of ethanol. Mixtures were applied onto Zymo Spin Columns to purify RNA (Direct-zol, RNA MiniPrep Plus, Cat #R2001; Zymo). RNA estimation, RT-PCR, and qPCR were performed based on a published protocol (Nolan et al., 2006). Primers for qPCR are listed below. To run qPCR, samples were preincubated at 95°C for 10 min, followed by 46 cycles of denaturation (95°C for 10 s), annealing (60°C for 30 s), and extension (72°C for 1 min). Samples were cooled at 40°C for 10 s before analysis. Kir4.1 and Cx43 levels normalized to HPRT1 level were calculated based on the $2^{-\Delta\Delta C(T)}$ method (Livak and Schmittgen, 2001). All values from one set of experiments were normalized to the average value of CTL before statistical analyses. Averages, SEMs, P values, and statistical tests are documented in Table S2. Animal numbers are documented at the end of the legend for Fig. 3. Primers for the detection of Kir4.1 are forward: 5′-AGTCTTGGCCCTGCCTGT-3′ and reverse: 5′-AGCGACCGACGTCATCTT-3′. Primers for Cx43 are forward: 5′-TCCTTTGACTTCAGCCTCCA-3′ and reverse: 5′-CCATGTCTGGGCACCTCT-3′, and the ones for HPRT1 are forward: 5′-GCTTGCTGGTGAAAAGGACCTCTCGAAG-3′ and reverse: 5′-CCCTGAAGTACTCATTATAGTCAAGGGCAT-3′.

## In vitro ubiquitination assay

For in vitro ubiquitination assays, recombinant mouse E1 (pET28-mE1), E2 (pGEX4T-1-UbcH5b), and E3s (pDEST-Nedd4-2) were expressed in BL21 Rosetta 2 (DE3) *Escherichia coli* and purified according to published protocols (Albert et al., 2002; Carvalho et al., 2012; Persaud et al., 2009) with slight modifications. pET28-mE1 was a gift from Jorge Eduardo Azevedo (Instituto de Investigação e Inovação em Saúde (i3S), Universidade do Porto, Porto, Portugal; plasmid #32534; Addgene; https://n2t.net/addgene:32534; RRID:Addgene_32534). Reaction mixture containing 2 mM ATP, 1 µM His-Ubiquitin (Cat #U-530; Boston Biochem), 50 nM mouse E1 enzyme, 150 nM E2 enzyme (UbcH5b), 300 nM E3 enzyme (Nedd4-2 WT, S233L, or H515P) in reaction buffer (50 mM Tris-Cl pH 7.5 at 37°C, 100 mM NaCl, 5 mM $MgCl_2$, 1 mM DTT, 0.05% [wt/vol] Tween 20) was incubated at 37°C. Reactions were stopped by the addition of 1X Laemmli Buffer, and polyubiquitin chain formation was detected by Western blotting using the anti-ubiquitin antibody. Signals from secondary antibodies were acquired and quantified by an Odyssey Imaging System (LI-COR). For the study of the time course of in vitro ubiquitination reaction in Fig. 2 B, the signal at the molecular weight of ~9 kD (i.e., the signal from free ubiquitin) at each time point was expressed as the level relative to free ubiquitin at the time point of 0 s (Fig. 2 B, left panel). Signals from polyubiquitin chains detected by the anti-ubiquitin antibody were quantified from the top of the SDS-PAGE separating gel to the approximate molecular weight of 50 kD. The same size of the region of interest was applied for the quantification of one set of experiments. Given that the level of free ubiquitin is close to 0% at 300 s in the left panel in Fig. 2 B, signals from polyubiquitin chains at each time point were expressed as the level relative to the time point of 300 s in the right panel of Fig. 2 B. For statistical analyses of free ubiquitin in the top panels of Fig. 2, D and F, signals from free ubiquitin were normalized to the Nedd4-2 level. The reduction of normalized signals at 150 s relative to the level at 0 s was expressed as depletion of free ubiquitin. For statistics of polyubiquitin chains in Fig. 2, D and F, signals from polyubiquitin chains were normalized to the level of Nedd4-2 first, and then further normalized to the average value of the control (N4-2 WT). Averages, SEMs, P values, and statistical tests are documented in Table S2. Sample numbers are documented at the end of the legend for Fig. 2. Experiments were done at least twice.

## Cell culture

HEK293FT cells were obtained from Thermo Fisher Scientific (Cat #R70007) and maintained in Dulbecco's Modified Eagle Medium (DMEM, Cat #41966-029; Thermo Fisher Scientific) supplemented by 10% Fetal Bovine Serum (FBS, Cat #10500-064; Thermo Fisher Scientific), 100 U/ml penicillin and streptomycin (Cat #15140-122; Thermo Fisher Scientific), and 2 mM glutamine (Cat #25030-024; Thermo Fisher Scientific) in 37°C incubator with 5% $CO_2$. For primary astrocyte culture, mutant mice backcrossed with the C57BL/6N >10 times were used. Cortical astrocyte culture was prepared according to previously published papers (Burgalossi et al., 2012).

## Lentivirus infection and dynasore treatment

Lentiviral vectors were produced in HEK293FT cells according to our previous publication (Hsia et al., 2014). Viruses were infected 16–24 h after plating astrocytes, which were harvested 5 d after infection. Viral titers were adjusted to express either recombinant N4-2 WT or the N4-2 C/S at the level comparable with the endogenous protein level (compare first, third, and fourth lanes in second Western blotting panels in Fig. 4 G). Under these conditions, ~90% of astrocytes were transduced. 100 µM dynasore was applied to astrocytes 40–48 h after plating astrocytes. Cells were harvested 12 h after the application of dynasore. Astrocytes were harvested in Laemmli buffer containing 5 mM $MgCl_2$ and 16.7 U/µl of Benzonase.

## In vivo ubiquitination assay

To show the ubiquitination of substrates in vivo, a cell-based ubiquitination assay was performed using HEK293FT cells as previously reported (Kawabe et al., 2010; Staub et al., 1997). After 48 h from transfection of HA-tagged Kir4.1 or Cx43 (Smyth and Shaw, 2013) with or without E3s, HEK293FT cells were washed with PBS containing 10 mM NEM, harvested in 200 µl of ubiquitination buffer (50 mM Tris-Cl pH 7.5 at 4°C, 300 mM NaCl, 0.2 mM PMSF, 1 µg/ml Aprotinin, 0.5 µg/ml Leupeptin, 10 mM NEM) containing 1% SDS, and incubated at 65°C for 20 min to denature proteins. To adjust the buffer suited for immunoprecipitation, SDS was diluted with 1.8 ml of ubiquitination buffer with 1% Triton X-100. The cell lysate was cleared by centrifugation at 10,000 rpm for 10 min at 4°C. Substrates were immunoprecipitated using anti-HA antibody coupled beads (Cat #A2095-1ML; Sigma-Aldrich). After washing beads three times with ubiquitination buffer with 1% Triton X-100, proteins on beads were eluted with Laemmli buffer and subjected to SDS-PAGE and Western blotting. pcDNA3.2-Cx43-HA was a gift from Robin Shaw (The Nora Eccles Harrison CVRTI, The University of Utah, Salt Lake City, UT, USA; plasmid #49851; Addgene; https://n2t.net/addgene:49851; RRID:Addgene_49851). For the ubiquitination assay using primary cultured astrocytes, primary cultured cortical astrocytes were prepared from N4-1/2 bDKO and N4-1/2 CTL mice at the age of postnatal day 0 (P0) or 2. Astrocytes were infected with HA-tagged Kir4.1- or Cx43-expression lentivirus. 5 d after infection, cultured astrocytes were washed with PBS containing 10 mM *N*-ethylmaleimide at least five times. Subsequent procedures were performed with the same protocol as the in vivo ubiquitination assay using HEK293FT cells. Experiments were performed at least twice.

## In vitro binding assay

pGex6P-1 plasmids that express truncated mutants of mouse Nedd4-2 (GenBank accession no. NM_001114386) encode amino acid sequences of Nedd4-2 as listed below:

(1) pGex6P-1 Nedd4-2 C2+WW1 (1–326 aa)
(2) pGex6P-1 Nedd4-2 WW1 (141–326 aa)
(3) pGex6P-1 Nedd4-2 WW1+WW2 (141–475 aa)
(4) pGex6P-1 Nedd4-2 WW2 (321–475 aa)
(5) pGex6P-1 Nedd4-2 WW3+WW4 (474–595 aa)
(6) pGex6P-1 Nedd4-2 WW2+WW3+WW4 (321–595 aa)

Proteins were expressed and purified from BL21 Rosetta 2 (DE3) *E. coli* cells, and bindings of Kir4.1-HA and Cx43-HA were studied as previously reported (Kawabe et al., 2010).

## Patch clamp recordings in astrocytes

Brain slices with 300 µm thickness were prepared for extracellular recordings. Astrocytes were labeled with sulforhodamine 101 (1 µM in ACSF, 34°C, 20 min) followed by washing with ACSF (10 min, 34°C). Later, brain slices were kept at room temperature until recordings. For whole-cell recordings, slices were transferred to a custom-made recording chamber on an upright microscope (BX51W1; Olympus). Sulforhodamine-labeled astrocytes were identified using epiflurorescence illumination (U-RFL-T; Olympus). Once cells were identified, astrocytes were patched with glass pipettes (4–6 MΩ) filled with intracellular solution (100 mM KCl, 50 mM K-gluconate, 10 mM HEPES, 0.1 mM EGTA, 0.3 mM GTP, 4 mM Mg-ATP, 0.4% Biocytin, pH 7.4) under transmitted light (TH4-200; Olympus, AxioCam MRm, Zeiss). Whole-cell voltage-clamp recordings were obtained using EPC10 amplifier (HEKA System). Membrane currents were low-pass filtered at 2 kHz and digitized at 20 kHz. Astrocytes were voltage-clamped to −80 mV and voltage–current plots were obtained using 10 mV voltage steps ranging from −120 to +40 mV or from −100 to +40 mV. For the rescue experiments, 30 µM fluoxetine or 100 µM barium chloride was included in ACSF during the recording.

## FRAP experiments

Astrocytes were plated on a glass bottom culture dish (Cat #80427; ibidi) in an astrocyte culture medium. An Okolab microscope incubator maintained the humidity, temperature, and $CO_2$ concentration of the cultures at 95–98%, 37°C, and 5%, respectively during FRAP imaging experiments. For the purpose of labeling, astrocytes were incubated with calcein-AM (Cat #17783-1MG; Sigma-Aldrich) for 30 min to 1 h prior to imaging to allow cells to uptake the fluorophore. After removing the calcein-AM-containing medium and washing cells with fresh medium three times, calcein in the astrocytes was bleached and images were obtained at 100-ms intervals in the first 15 s after bleaching and subsequently at 5-s intervals with an inverted Nikon Eclipse Ti microscope equipped with hardware autofocus (Perfect Focus), a spinning disk confocal unit (CSU-W1 Yokogawa), and an Andor FRAPPA unit used for bleaching. Fluorescence from the cultured astrocytes was observed with a 40× oil immersion objective lens (1.30 NA) and captured with an Andor iXon Ultra 888 EMCCD camera. At each time point (t), the background signal intensities [$I_{back}(t)$] and photobleached or non-bleached reference cell intensities [$I_{frap}(t)$ or $I_{ref}(t)$] were extracted with the NIS Elements (v. 4.51) software package during imaging and then analyzed with Microsoft Excel. The bleached time point is defined as tb. The sampling number at the bleach time point is defined as $n_{bleach}$. FRAP was estimated as follows;

(1) Normalized signal intensity from beached cells (Infrap(t))

$$\mathrm{Infrap}(t) = \left(I_{frap}(t) - I_{back}(t)\right) / \left(I_{ref}(t) - I_{back}(t)\right)$$

(2) Full range of bleaching ($I_{fullbleach}$)

$$I_{fullbleach} = \Sigma_{t=0}^{t=tb-1} \mathrm{Infrap}(t) / \mathrm{nbleach} - \mathrm{Infrap}(tb)$$

(3) FRAP efficiency (%) at t = $(\mathrm{Infrap}(t) - \mathrm{Infrap}(tb)) / I_{fullbleach} \times 100$

Averages, SEMs, P values, and statistical tests are documented in Table S2. Sample numbers are documented at the end of the legend for Fig. 6. Experiments were performed at least twice by an observer blinded to genotypes.

### In vivo recording of γ-oscillations

5-wk-old mice were anesthetized with urethane (1.3–1.5 mg/kg) and placed on a stereotaxic frame at 36°C (ATC1000, Cat #61805; WPI) while monitoring. A craniotomy of 1 × 1 mm was made centered at 2 mm posterior from lambda and 2.6 mm lateral from the midline. A linear multielectrode array (Cat #A1-16-3mm-50-177; Neuronexus) coated with DiI was inserted from the cortical surface orthogonally and placed at 2.5 mm from the brain surface to target CA3. Signals from field potentials were preamplified (Cat #HS-18; Neuralynx) and sent to an acquisition board (Digital Lynx 4SX, Cat #DL-4SX; Neuralynx) at a sampling rate of 32 kHz. After 30 min of recording the baseline activity, kainate (20 mg/kg, Cat #BN0281; BioTrend) was injected intraperitoneally and the recordings were extended for 60 min. After recordings, the brain was fixed in 4% paraformaldehyde. The position of the electrode was validated based on signals from DiI in the brain tissue after sectioning.

LFPs were analyzed in MATLAB (Mathworks). The signal was filtered (0.7–400 Hz) and downsampled to 1 kHz. Power was calculated with the multitaper method (Chronux Package, https://chronux.org, five tapers, time bandwidth product of 3 and 0.2 s overlap). The data were normalized by dividing the power spectra of each individual time point after kainate injection by the average power spectra of the baseline activity. The range of the γ-band was 25–45 Hz. Values between groups across time were compared using repeated measures two-way ANOVA (genotype × time), followed by post-hoc Fisher's least significant difference corrections. Averages, SEMs, P values, and statistical tests are documented in Table S2. Sample numbers are documented in the figure legend for Fig. 9.

### Statistical analysis

For statistical analyses, Graph Pad Prism 5 or MATLAB software was used. All results were expressed as mean ± SEM. Two independent groups were analyzed by Student's or nested two-tailed *t* test, where significance was defined as P < 0.05. For electrophysiological experiments, data were acquired by observers blinded to genotype except for the rescue experiment in Fig. 5, Fig. 7, H–M, and Fig. 8, where we applied fluoxetine or GAP26 only to N4-2 bKO or AstKO samples. One-way ANOVA test was applied for the comparison of one parameter between three or more groups. When a relatively small difference was expected (i.e., partial rescues of reduced γ-oscillation in Nedd4-2 AstKO by GAP26 or fluoxetin), Newman–Keuls post-hoc test was applied for the comparison of the two groups. Otherwise, Tukey post-hoc test was used. For comparisons of more than two parameters between two groups in Fig. 9, a two-way ANOVA with LSD post-hoc test was applied.

### Quantification of electrophysiological recording using acute slices

For field recording experiments, absolute powers and frequencies of γ-oscillations were analyzed statistically. For patch clamp experiments, absolute membrane currents and resistances were analyzed. All sets of experiments were performed twice with two or more pairs of animals. Outliers were excluded from statistical analyses using Graph Pad Prism software. Datasets with a serial decline in the baseline of the LFP were also excluded from analyses. Averages, SEMs, P values, and statistical tests are documented in Table S2. Numbers of recorded cells and acute slices are documented within legends of Figs. 1, 5, 7, and 8.

### Quantitative immunohistochemistry

5-wk old mice were perfused intracardially with 4% paraformaldehyde and the brains were dissected overnight after fixation with 4% paraformaldehyde at 4°C. After washing the brains with PBS several times, they were cryoprotected with 30% sucrose. 15 μm hippocampal sections were prepared using a cryostat (CM3050 S; Leica) and were stained with primary and secondary antibodies. Signals from secondary antibodies were acquired by Leica TCS-SP2 confocal microscopy using 20X oil immersion objective lens with the digital zoom of 1X. Averages, SEMs, P values, and statistical tests are documented in Table S2. Experiments were performed with two pairs of animals. Numbers of sections are documented in the legends of Fig. 7.

### Online supplemental material

Fig. S1 shows γ-oscillations in *Nedd4-1*^f/f^;*Nedd4-2*^f/f^;*NEX*-Cre (N4-1/2 nDKO) and *Nedd4-1*^f/f^;*EMX*-Cre (N4-1 bDKO). Fig. S2 shows the lack of impacts of fluoxetine on membrane currents and γ-oscillations in *Nedd4-2*^f/f^ (N4-2 CTL). The mass spectrometry proteomics data have been deposited to the ProteomeXchange Consortium via the PRIDE (Perez-Riverol et al., 2022) partner repository with the dataset identifier PXD046450. Other data are available from the corresponding author upon reasonable request. Table S1 shows details of proteome data, related to Fig. 3 E. Highlighted in magenta and green are proteins up- or downregulated in N4-1/2 bDKO for both forward and reverse experiments significantly. In the tab "all proteins simplified," information about all the identified proteins is provided. In the tab "all proteins raw data," original data of all identified proteins are listed. Table S2 shows details of statistical tests. Student's and nested *t* tests; related to Fig. 1, C, D, G, and H; Fig. 2, E and G; Fig. 3, C, H, I, K, M–O, Q, R, T, and U; Fig. 4 G; Fig. 6 E; and Fig. 7, E–G, O, and P. Details of multiple comparison tests; related to

Fig. 4, H and I; Fig. 5, E and F; Fig. 7, L and M; and Fig. 8, D, E, I, and J. Details of post-hoc tests in ANOVA tests; related to Fig. 4, H and I; Fig. 5, E and F; Fig. 7, L and M; and Fig. 8, D, E, I, and J. Details of repeated measures two-way ANOVA test; related to Fig. 9 C. Table S3 provides the antibody list.

## Acknowledgments

We thank Vincent O'Connor, Erinn Gideons, and James Daniel for their helpful comments and critical reviews of the manuscript, and Klaus Nave for his support. We are grateful to Fritz Benseler, Klaus-Peter Hellmann, Bernd Hesse-Niessen, Ivonne Thanhäuser, Dayana Warnecke, Christiane Harenberg, Maik Schlieper, Lars van Werven, Dörte Hesse, Monika Raabe, Uwe Plessmann, and Aiko Miyakawa for excellent technical support, and to the Animal Facilities of the Max Planck Institute for Multidisciplinary Sciences for the maintenance of mouse colonies and support for mass spectrometry. We thank Johannes Hirrlinger, Frank Kirchhoff, Nicola Strenzke, and Gulnara Yamanbaeva for their advice. We thank Sandra Goebbels (Max Planck Institute for Multidisciplinary Sciences, Göttingen, Germany), Kevin R. Jones (University of Colorado, Boulder, CO, USA), and Hongkui Zeng (Allen Institute for Brain Science, Seattle, WA, USA) for providing Nex-Cre, EMX-Cre, and ROSA26-tdTom mouse lines, respectively; Thorsten Bus and Rolf Sprengel (Max Planck Institute for Medical Research, Heidelberg, Germany) for providing brain tissue and tissue extracts of control and Gria1$^{-/-}$ mice; and Jorge E. Azevedo (Instituto de Investigação e Inovação em Saúde (i3S), Universidade do Porto, Porto, Portugal), David Baltimore (California Institute of Technology, Pasadena, CA, USA), Richard Huganir (Johns Hopkins University School of Medicine, Baltimore, MD, USA), Didier Trono (Swiss Federal Institute of Technology Lausanne (EPFL), Lausanne, Switzerland), Robin Shaw (The Nora Eccles Harrison CVRTI, The University of Utah, Salt Lake City, UT, USA), and Marc Timmers (German Cancer Consortium (DKTK), Freiburg, Germany) for providing plasmids.

This work was supported by Canadian Institutes of Health Research Foundation (D. Rotin), the German Research Foundation (SPP1365/KA3423/1-1 and KA3423/3-1, H. Kawabe; SPP1757/SA2114/2-1 and SA2114/2-2, G. Saher), the Fritz Thyssen Foundation (H. Kawabe), and the Japan Society for the Promotion of Science (JSPS) KAKENHI grant number JP20K06536 (K. Hanamura), JP22K07864 (N. Koganezawa), JP22H05526 (N. Koganezawa), JP15K21769 (H. Kawabe), JP20K07334 (H. Kawabe), JSPS Core-to-Core program (grant number JPJSCCA20220007) (H. Kawabe), and The Mother and Child Health Foundation (grant number 29-9, H. Kawabe; grant number R03-K1-2, N. Koganezawa), Ohsumi Frontier Science Foundation (H. Kawabe), the Takeda Science Foundation (N. Koganezawa and H. Kawabe), Daiichi Sankyo Foundation of Life Science (H. Kawabe), TERUMO Life Science Foundation (H. Kawabe), The Naito Foundation (H. Kawabe), and The Uehara Memorial Foundation (H. Kawabe). Open Access funding provided by the Max Planck Society.

Author contributions: B. Altas: conceptualization, project administration, investigation, validation, formal analysis, writing-original draft, and writing-review and editing; H.-J. Rhee, A. Ju, H.C. Solís, S. Karaca, J. Winchenbach, O. Kaplan Arabaci, M. Schwark, M.C. Ambrozkiewicz, C. Lee, L. Spieth, G.L. Wieser, V.K. Chaugule, I. Majoul, M.A. Hassan, R. Goel, and S.M. Wojcik: investigation, validation, formal analysis, and writing-review and editing; A. Pichler, M. Mitkovski, L. de Hoz, A. Poulopoulos, H. Urlaub, O. Jahn, G. Saher, N. Brose, and J. Rhee: conceptualization and writing-review and editing; N. Koganezawa and K. Hanamura: investigation, validation, formal analysis, writing-review and editing, and provided funding for the research; D. Rotin: conceptualization, writing-review and editing, and provided funding for the research; H. Kawabe: supervision, conceptualization, provided funding for the research, project administration, writing-review and editing.

Disclosures: The authors declare no competing interests exist.

Submitted: 14 February 2019

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

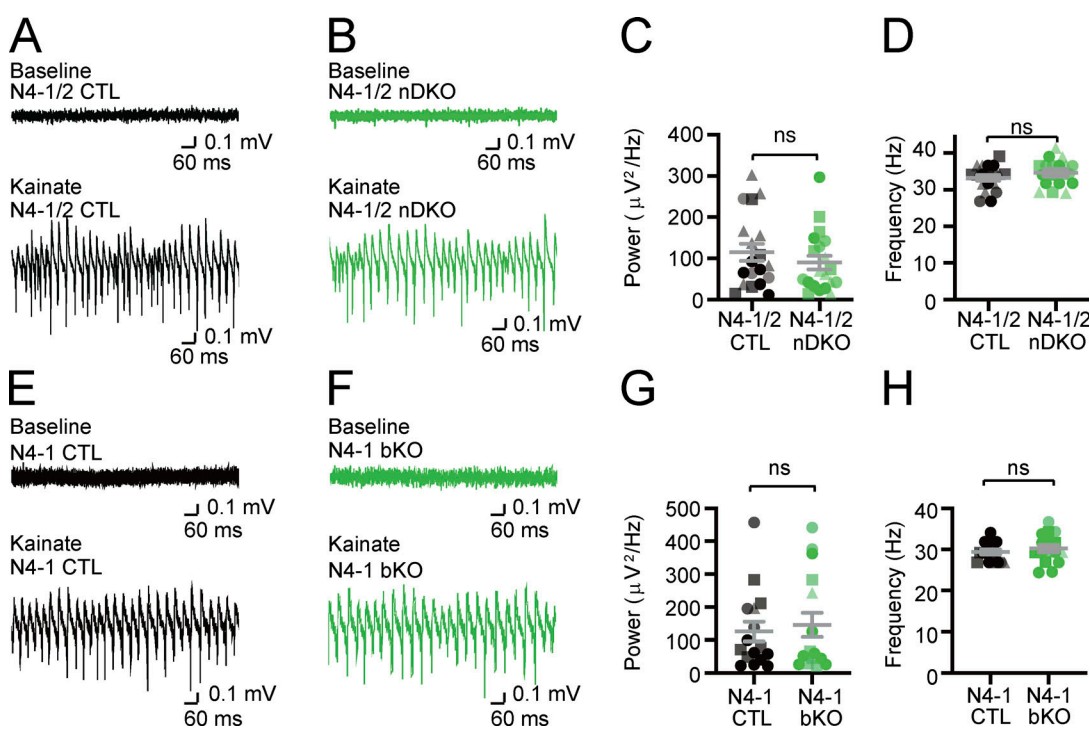

Figure S1. **Intact γ-oscillations in acute slices from Nedd4-1 and Nedd4-2 neuron-specific knockout (N4-1/2 nDKO) and Nedd4-1 brain-specific knockout (N4-1 bKO) mice. (A and B)** Representative recordings in CA3 hippocampal regions of acute brain slices from N4-1/2 CTL (A) and N4-1/2 nDKO (B) mice before (baseline) and during (Kainate) induction of γ-oscillations with 100 nM kainate. **(C and D)** Average powers (C) and frequencies (D) of γ-oscillations in N4-1/2 CTL (black dots) and N4-1/2 nDKO (green dots). **(E–H)** Intact power of γ-oscillations in *Nedd4-1*^f/f;*EMX*-Cre (N4-1 bKO) mice. Representative recordings in CA3 region of hippocampus slices from *Nedd4-1*^f/f (N4-1 CTL) (E) and N4-1 bKO (F) mice before (Baseline) and during (Kainate) the application of 100 nM kainate. Average powers (G) and frequencies (H) of γ-oscillations in N4-1 CTL (black dots) and N4-1 bKO (green dots). Different tones and shapes of dots in dot plots in C, D, G, and H represents data from different mice. Numbers of recorded slices (*n*) and mice (*N*); (C and D), *n* = 19 and *N* = 4 for N4-1/2 CTL, *n* = 20 and *N* = 4 for N4-1/2 nDKO; (G and H), *n* = 16 and *N* = 4 for N4-1 CTL, *n* = 16 and *N* = 4 for N4-1 bKO. Results are shown as mean ± SEM. ns, 0.05 < P (two-tailed Student's *t* test). Data distribution was assumed to be normal, but this was not formally tested. See also Table S2.

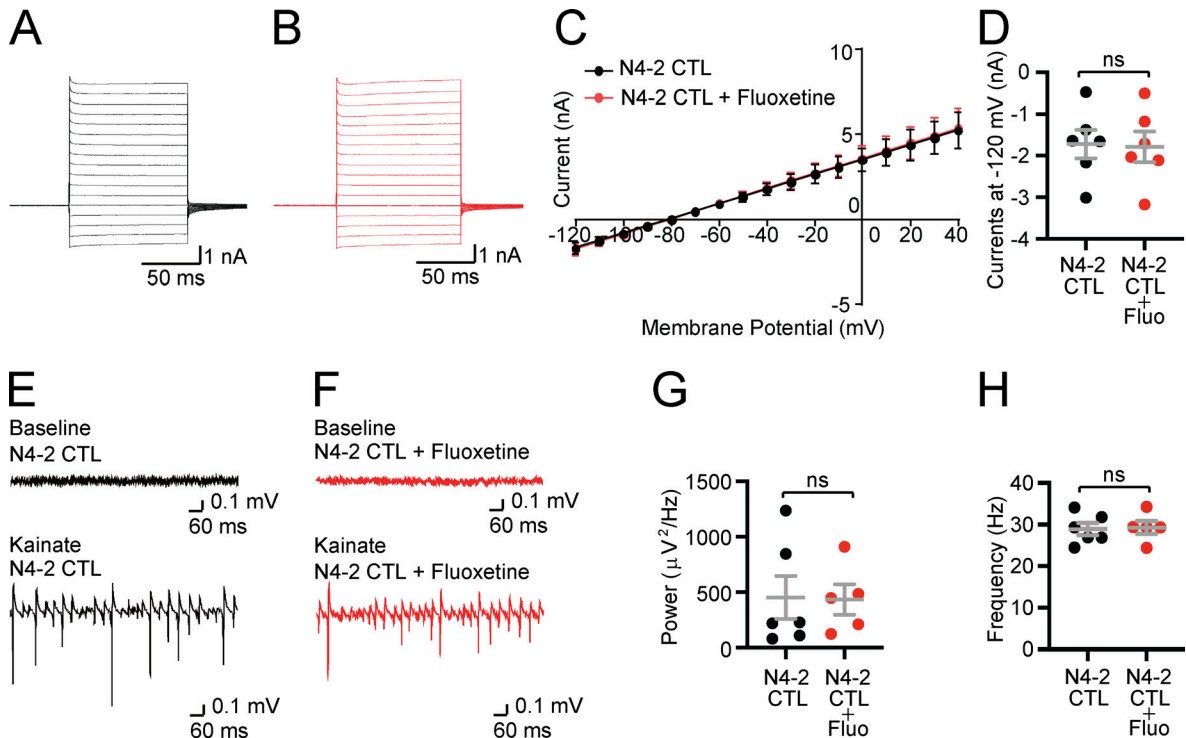

Figure S2. **The lack of impacts of fluoxetine on membrane currents and γ-oscillations in *Nedd4-2*^f/f (N4-2 CTL). (A and B)** Example current traces in N4-2 CTL astrocytes without (A) and with fluoxetine (B). Cells were voltage clamped at −80 and +10 mV voltage steps were applied from −120 to +40 mV during recording currents. **(C and D)** Quantifications of membrane currents in N4-2 CTL without and with fluoxetine. **(C)** Voltage–current plots from N4-2 CTL (black trace) and N4-2 CTL with fluoxetine (red trace). **(D)** The current at the membrane potential of −120 mV showed no effects of fluoxetine in N4-2 CTL. **(E and F)** Representative recordings in CA3 hippocampal regions of acute brain slices from N4-2 CTL (E) and N4-2 CTL with fluoxetine (F) mice before (baseline) and during (Kainate) induction of γ-oscillations with 100 nM kainate. **(G and H)** Average powers (G) and frequencies (H) of γ-oscillations in N4-2 CTL (black dots) and N4-2 CTL with fluoxetine (red dots). Results are shown as mean ± SEM. Numbers of recorded cells (*n*) in D; *n* = 6 for N4-2 CTL, *n* = 6 for N4-2 CTL + Fluo. Numbers of recorded slices (*n*); (G and H), *n* = 6 for N4-2 CTL, *n* = 5 for N4-2 CTL + Fluo. Results are shown as mean ± SEM. ns, 0.05 < P (two-tailed Student's *t* test). Data distribution was assumed to be normal, but this was not formally tested. See also Table S2.

**Provided online are Table S1, Table S2, and Table S3. Table S1 shows details of proteome data. Table S2 shows details of statistical tests. Table S3 provides the antibody list.**

