## [Peer Review File · The Journal of Cell Biology]

Nedd4-2-dependent regulation of astrocytic Kir4.1 and Connexin43 controls neuronal network activity

Bekir Altas, Hong Jun Rhee, Anes Ju, Hugo Solís, Samir Karaca, Jan Winchenbach, Oykum Kaplan-Arabaci, Manuela Schwark, Mateusz Ambrozkiewicz, ChungKu Lee, Lena Spieth, Georg Wieser, Viduth Chaugule, Irina Majoul, Mohamed A. Hassan, Rashi Goel, Sonja Wojcik, Noriko Koganezawa, Kenji Hanamura, Daniela Rotin, Andrea Pichler, Miso Mitkovski, Livia de Hoz, Alexandros Pouloupoulos, Henning Urlaub, Olaf Jahn, Gesine Saher, Nils Brose, JeongSeop Rhee, and Hiroshi Kawabe

Corresponding Author(s): Hiroshi Kawabe, Gunma University

Review Timeline:

Submission Date:	2019-02-14
Editorial Decision:	2019-03-26
Revision Received:	2021-10-21
Editorial Decision:	2021-11-19
Revision Received:	2023-09-04
Editorial Decision:	2023-09-22
Revision Received:	2023-10-31

Monitoring Editor: Louis Reichardt

Scientific Editor: Tim Spencer

Transaction Report:

DOI: <https://doi.org/10.1083/jcb.201902050>

March 26, 2019

Re: JCB manuscript #201902050

Dr. Hiroshi Kawabe
Max Planck Institute of Experimental Medicine
Department of Molecular Neurobiology
Hermann Rein Str. 3D
Göttingen 37075
Germany

Dear Dr. Kawabe,

Thank you for submitting your manuscript entitled "Physiological and pathophysiological homeostasis of astroglial channel proteins by Nedd4-2". The manuscript was assessed by expert reviewers, whose comments are appended to this letter. We sincerely apologize for the delay in communicating our decision to you and thank you for your patience with the peer review process. We invite you to submit a revision if you can address the reviewers' key concerns, as outlined here.

Overall, each of the reviewers finds your work interesting and clearly competitive, following adequate revision, for acceptance by this journal. We have read each of the reviewer's comments with your manuscript in front of me. Overall, we think that addressing them carefully will clearly enhance the quality and impact of your study.

More specifically, while we expect you to provide a response to each item listed in the different assessments provided by each reviewer, we want to call your attention to some items that appear to us to be of special importance.

Regarding comments of reviewer #1, I would especially call your attention to items listed under the headings: Figure 2D, F; the paragraph following the heading "p.13"; Figure 5 and Figure 8.

Concerning those of reviewer #2, I consider it especially important to address items #2, 3, 4, 5, 6 and 7.

Concerning those of reviewer #3, major points 1, 2, 3, 4, 6 and 8 seem essential to address plus minor comments 4, 5, and 6.

To summarize, each of the three reviewers is quite supportive of publication in this journal of a revised manuscript. It is important, however, that you take the time and perform the experiments needed to address convincingly their scientific concerns. Additionally, please pay special attention to the areas of the manuscript where reviewers felt that the clarity of your logic and experimental design could be improved.

Assuming that you are willing to undertake the effort needed to address the comments of each reviewer, please include with a revision a letter describing your responses and changes made in response to each specific concern of the reviewers. Additionally, we would like to request that you provide a copy of your manuscript with changes highlighted to facilitate the review of your text.

While you are revising your manuscript, please also attend to the following editorial points to help expedite the publication of your manuscript. Please direct any editorial questions to the journal office. Please let us know if you have any questions about the revision process or would like to discuss these points further.

GENERAL GUIDELINES:

Text limits: Character count for an Article is < 40,000, not including spaces. Count includes title page, abstract, introduction, results, discussion, acknowledgments, and figure legends. Count does not include materials and methods, references, tables, or supplemental legends.

Figures: Articles may have up to 10 main text figures. Figures must be prepared according to the policies outlined in our Instructions to Authors, under Data Presentation, <http://jcb.rupress.org/site/misc/ifora.xhtml>. All figures in accepted manuscripts will be screened prior to publication.

Supplemental information: There are strict limits on the allowable amount of supplemental data. Articles may have up to 5

supplemental figures. Up to 10 supplemental videos or flash animations are allowed. A summary of all supplemental material should appear at the end of the Materials and methods section.

The typical timeframe for revisions is three months; if submitted within this timeframe, novelty will not be reassessed at the final decision. Please note that papers are generally considered through only one revision cycle, so any revised manuscript will likely be either accepted or rejected.

Thank you for this interesting contribution to the Journal of Cell Biology. You can contact us at the journal office with any questions, cellbio@rockefeller.edu or call (212) 327-8588.

Sincerely,

Louis Reichardt, PhD
Editor, Journal of Cell Biology

Melina Casadio, PhD
Senior Scientific Editor, Journal of Cell Biology

Reviewer #1 (Comments to the Authors (Required)):

Here Atlas et al. address the basis for neural network regulation by Nedd4-2, a ubiquitin E3 ligase whose mutations are associated with epilepsy. The authors find that brain-specific loss of Nedd4-2 alters the power of gamma oscillations in hippocampal area CA3. They then go on to identify that astrocyte Kir4.1 and Connexin 43 are Nedd4-2 targets, whose protein levels are kept in check by Nedd4-2-mediated ubiquitination. Altered expression levels of Kir4.1 and Connexin 43 by dysregulation of Nedd4-2 is suggested to affect synchronized network activity and contribute to epilepsy. The study makes use of a variety of experimental approaches, from in vitro and in vivo electrophysiology, biochemistry to FRAP measurements, and the experiments have been carefully performed. The main findings are important and should be of interest to a broad readership. However, I have some minor comments that require consideration:

The writing, at times can be problematic, in that the results are overstated or overinterpreted, and some explanations are oversimplified. The text will benefit from careful editing.

Page 5, line 7. "All of them play roles in ion and solute exchange..." "All of them" is too vague. What are they? It might be helpful to give specific examples.

Page 5, line 13-14. "Its conditional knock-out (KO) in mice..." What is conditional here? (e.g. region-specific, cell-type specific, developmental expression, etc.)

Page 5, line 9. "Extracellular K⁺ is taken up by astrocytes is quickly dissipated within a gap-junction connected astrocyte network." Reference is missing.

Page 5, last sentence. It is not clear how this statement is related to K⁺ uptake by Kir4.1 channels.

Figure 2D,F. Top panels - In order to avoid confusion and maintain consistency with panel B (left), the plots here should also compare the relative free ubiquitin rather than depletion of free ubiquitin.

Figure 3B, p.9. A lack of change in the protein level of GluA1 is not a sufficient evidence for ruling out the possible contribution of altered AMPA receptor activity to the reduced power of gamma oscillations. Synaptic surface level of AMPA receptors needs to be examined rather than the total receptor content, and more importantly, GluA2-containing AMPA receptors could be more relevant in influencing basal network activity. The authors should rephrase this statement.

p.13. "Cultured astrocytes were prepared from the cortex but not hippocampus because of homogeneous expression patterns of Cx43 and N4-2 throughout the cortex and hippocampus." The logic is not clear here. Perhaps the authors wish to justify the use of cortical astrocytes rather than hippocampal astrocytes, despite the fact that oscillation experiments have been performed in hippocampal slices - is this correct? The sentence needs to be clarified to convey the intention that there is an assumption here that cortical astrocytes should behave in a similar way as hippocampal astrocytes given the similar expression levels of Cx43 and N4-2 between the two brain regions.

Figure 5 is disjointed from the flow of the manuscript. Does fluoxetine affect the I-V curve in control astrocytes? Does dynasore treatment in controls shift the I-V curve relative to mutant astrocytes or the mutant + dynasore condition? As it stands, the figure might be best included as supplementary information.

Figure 7N-P. It is not just the number of PV interneurons but it is their synaptic inputs and outputs that affect the CA3 circuit dynamics.

Figure 8. Does fluoxetine or GAP26 cause a further increase in the power of gamma oscillations in control slices?

Page 16, line 2. Connexins are not generally defined as ion channels as they also permeate larger molecules.

Reviewer #2 (Comments to the Authors (Required)):

In this manuscript, Atlas and colleagues describe a mechanism through which E3 ubiquitin ligase Nedd4-2 in astrocytes regulate γ -oscillatory neuronal network activity. They show that Nedd4-2 controls the degradation of ion channels Kir4.1 and Connexin43 in astrocytes. Using a combination of mouse genetics, proteomics, in vitro and in vivo biochemical analyses and ubiquitination assays, the authors found that loss of Nedd4-2 results in a significant increase in the protein levels of both Kir4.1 and Connexin43. The presence of Nedd4-2 increases the levels of K63 ubiquitination on Kir4.1 and Connexin43, targeting these proteins for degradation via the endo-lysosomal pathway. Physiologically, loss of Nedd4-2 in astrocytes increases membrane conductance in a Kir4.1-dependent manner and increases gap junction coupling in vitro. Nedd4-2 astrocyte knockout animals show reduced gamma-oscillations in response to kainate treatment, which can be partially rescued with pharmacological inhibition of Kir4.1 or Connexin43. Mutations in Nedd4-2 are found in familial epilepsy, and the authors' data suggests that these mutations result in Nedd4-2 gain of function resulting in faulty ion buffering in astrocytes.

In sum, this is an exciting study that potentially uncovers a novel molecular mechanism in astrocytes which is potentially relevant to epilepsy. However, I have a number of concerns that the authors should address thoroughly to clarify critical several points, justify their interpretation of some of the data and provide validation for some of the methods.

1. The manuscript starts out by using Nedd4-1 and Nedd4-2 double knockouts, including the initial screen which identified Kir4.1 and Cx43. The comparison is made to a CTL (control ? group) of unknown genotype(s) and backgrounds. The authors should clearly state which genotypes were used as controls.
2. Related to 1, there is also strong concern that the authors may have failed to completely knockout Nedd4-1 in their double KO model since there are faint bands in Figure 3B and F when probing for Nedd4-1 protein expression. Please explain.
3. The authors use Eaat2 protein expression as an indirect indicator of astrocyte number. This is not an accurate analysis, and an actual cell count by staining with astrocyte markers in the CA3 is required to fully make the claim that astrocyte numbers did not change.
4. There is a sharp difference in the power of γ -oscillations when comparing N4-2 KO (1G) and N4-2 astrocyte specific KO (Figure 8D and 8I), suggesting that there might be a non-astrocyte contribution to neuron network synchronicity i.e. a neuronal component this was not discussed at all. Please explain.
5. The authors suggest that N4-1 compensates for N4-2 without any experimental proof. This may be OK to include in discussion as a possibility but should be omitted from the results.
6. The authors suggest that ubiquitination by N4-2 activates the lysosomal pathway, resulting in endocytosis of Kir4.1 and Cx43. However, they fail to explain for the seemingly additive effect on Cx43 protein expression (Figure 4H) when both lysosome activity and ubiquitination is blocked, which suggest that N4-2 mediated endocytosis is not the only factor regulating Cx43 cell surface expression.
7. Finally, there was only a partial rescue in the power of γ -oscillation when N4-2 KO astrocytes were treated with fluoxetine (Figure 8D). Since the authors argue that fluoxetine is a specific inhibitor of Kir4.1, this suggests that dysregulation of K⁺ buffering is not the only factor influencing neuronal network activity. Maybe Kir4.1 and CX43 work in paralele rather than in tandem. The authors should comment on this and potentially perform an experiment where both Cx43 and Kir4.1 are blocked.

Minor points

1. In Figure 4A, the authors should re-run the samples in the correct order for the figure so that there is no cut-and-paste involved.
2. In Figure 7D-G, where does the sample come from for Western blot? Is it from hippocampal lysate, cortical, or whole brain?
3. In Figure 7, the authors indicate that 5% of TdTomato+ cells are not positive for S100beta, indicating that a cell type other than astrocytes is undergoing Cre-mediated recombination. Since the authors are using a tamoxifen dosing timeline which differs from previously published studies (in which tamoxifen was administered in the adult), they should determine whether any NeuN+ (and in particular PV+) cells are expressing TdTomato.
4. The authors make an incorrect statement in the discussion section on p.19. Cx43 in astrocytes pairs with Cx47 on oligodendrocytes, not with Cx32 as the authors have stated.
5. In the methods section, please clarify the method used for Quantitative Western blotting. The description of methods for the In

vitro Ubiquitination Assay indicated that fluorescent secondary antibodies and the Odyssey Imaging System were used, but it is unclear for the blots in other figures. Was this same method of quantification used?

Reviewer #3 (Comments to the Authors (Required)):

In the manuscript entitled 'Physiological and pathophysiological homeostasis of astroglial channel proteins by Nedd4-2', Altas and colleagues have demonstrated that a ubiquitin E3 ligase Nedd4-2 ubiquitinates and downregulates expression of Kir4.1 and Cx43 in astrocytes, leading to increase in the membrane ion permeability and in gap junction connectivity. In general, the manuscript is well written but there are some important issues, which need to be solved to support their conclusion.

Major points:

1. Figure 1. as used in Figure 3M-R, it is critical to show gamma-Oscillations using the N4-1bKO samples to support their conclusion.
2. in Figure 2, what is the reason why E271A is not tested for its activity in vitro? It would be important to test this mutant to strengthen the statement of authors. Based on the method section, used samples are not denatured before IP. This is absolutely critical to conclude if Cx43 is ubiquitinated or not, especially because HA-IP, HA-Blt samples seem not to show modification of HA-Cx43.
3. Following up the above point #2, for all the in cell ubiquitination assays, it is essential to denature once the lysates before IP. (after denaturing, one needs to adjust the buffer suited for IP)
4. Figure 4B: Regarding authors' description as 'We interpret this data as compensatory homeostatic regulation of Cx43-HA by K63-ubiquitination-dependent lysosomal degradation.', treatment of cells with Bafilomycin A1 would clarify this point. Since it is rather simple experiments and will help authors' conclusion, it is important to test BafA1-treated samples.
5. Figure 3H: Especially because evidence of Dynamin inhibition in these samples are not shown, and because a more recent study has shown about Dynamin-independent roles of Dynasore (PMID: 29150487 DOI:10.1242/jcs.211755), it is better to rephrase the following sentence; To study whether N4-2 promotes the degradation of Kir4.1 and Cx43 via the endolysosomal pathway, clathrin-mediated endocytosis was blocked with dynasore, a specific blocker of dynamin (Macia et al., 2006) in primary astrocyte cultures (Figure 4H).
6. it is very difficult to understand what kind of samples are used if it is written only as 'samples'. For example, it is unclear what were analysed in Figure 3I and J from the figure legends nor the text. Thus, it is very hard for readers to understand if the conclusion made by authors such as 'controls their surface expression (page 16 line 3)' is supported by the provided data or not. Please fix.
7. in the Discussion section: page 16, line 2: The authors state that a single E3 ligase ubiquitinates XXXX, but based on the data in Figure 4C and D, and as authors interpreted these data sets as 'probably by other E3s involved', it is very confusing where this statement came from.
8. Figure 2 and 10: N4-2 mutants S233L or H515P forms more free ubiquitin chains in vitro but no evidence provided if Kir4.1 and Connexin43 are more ubiquitinated. Without examining the ubiquitination status of the substrates by these mutants at least in vitro, it is not possible to conclude if the effect on the protein expression levels of the substrates are direct or indirect. Even though Fig 10 is just a schema, it is suggested to perform such experiments to test one of the main points of this study.

Minor points:

1. in Figure 2, to obtain better ideas where the three human mutations of N4-2 are localized, it is recommended to add a similar schematic as Fig 4E and indicate the mutation sites.
2. in the Method section, Quantitative Western blotting: It will be very helpful to comment one phrase that the Odyssey system was used as this is critical info (although which is properly indicated in the Hsia et al paper).
3. page 9, line 8-9: the statement 'N4-1 and N4-2 generate K63-linked polyUbi chains (Maspero et al., 2013), which are involved in the endocytosis and lysosomal degradation of transmembrane proteins (Acconcia et al., 2009).' is not entirely correct. As Acconcia et al wrote in their review, only some specific cases like AMSH-dependent EGFR degradation is controlled by K63 chains. Please rephrase.
4. page 9, Data not shown. It is highly recommended to add the data possibly as a supplementary figure.
5. page 17, line 3-5, please indicate at least citations for this statement ' Nevertheless, the major consequences of loss-of function of a specific E3 in a given cell type are often due to aberrant ubiquitination of only a few substrates, indicating a substantial cell-type specificity and selectivity of E3-substrate interactions.' since this concept seems not to be generalized in the research field.
6. page 17, regarding the statement 'The present study demonstrates that a dominant function of N4-2 in the adult brain is to SPECIFICALLY and SELECTIVELY ubiquitinate and downregulate Kir4.1 and Cx43 in astrocytes.': based on the data in 3E, there are more targets shown in double KO samples of N4-1 and 2. The chosen targets as Kir4.1 and Cx43 in this study are indeed the N4-2 substrates, however, it is unclear if the above statement can be made based on the shown data in this study.
7. please fix the qRT-PCR method section from the protocol style to the method style.

Resubmission of the revised manuscript 'Physiological and pathophysiological homeostasis of astroglial channel proteins by Nedd4-2' by Altas et al. (JCB manuscript #201902050)

**Point-by-point response to the reviewers' comments
(Reviewers' comments in bold)**

Please allow us to first explain the substantial delay in returning the revised MS. In 2019 after completing PhD thesis defence, the first author of the manuscript, Dr. Altas Bekir moved from the Max Planck Institute of Experimental Medicine to Maryland University to start his new project as a postdoctoral researcher. I myself moved to Gunma University in Japan as a full professor. An unusual duration for a manuscript revision is due to the tri-continental nature of our collaboration, which includes sites in Japan, Germany, and the United States. In the individual circumstances we each faced in dealing with the pandemic and staggered shutdowns of our institutions. Thanks to the time afforded, I am proud to say that we have managed this significant and rigorous revision of our findings by addressing your requests. We all hope that you share our enthusiasm to manage the revision of our manuscript under such a difficult situation.

Reviewer #1

[The study makes use of a variety of experimental approaches, from in vitro and in vivo electrophysiology, biochemistry to FRAP measurements, and the experiments have been carefully performed. The main findings are important and should be of interest to a broad readership.]

We are very happy that this reviewer finds that the experiments were carefully performed and the main findings are important and should be of interest to a broad readership.

[The writing, at times can be problematic, in that the results are overstated or overinterpreted, and some explanations are oversimplified. The text will benefit from careful editing.]

We appreciate this suggestion. We changed the manuscript based on this reviewer's major and minor comments, as seen below.

[Page 5, line 7. "All of them play roles in ion and solute exchange..." "All of them" is too vague. What are they? It might be helpful to give specific examples.]

We thank this reviewer for pointing this out. We included some examples for ion channels and ion pumps. Regarding the gap junctions, we did not include examples because Connexin43 is the most dominant Connexin to form gap junctions in astrocytes and because it is explained in the following parts of the manuscript.

[Page 5, line 13-14. "Its conditional knock-out (KO) in mice..." What is conditional here? (e.g. region-specific, cell-type specific, developmental expression, etc.)]

We changed this part of the manuscript from 'Its conditional knock-out (KO) in mice...' to 'Its brain-specific conditional knock-out (KO) in mice...'. In Djukic et al., 2007, where Kir4.1 conditional KO was characterized, the GFAP promoter was used to drive Cre expression. The authors in that paper discussed possible GFAP promoter activity in progenitor cells (Malatesta et al., *Development* 127: 5253-5263, 2000; Malatesta et al., *Neuron* 37: 751-764, 2003; Casper and McCarthy, *Mol Cell Neurosci.* 31: 676-84, 2006). We feel it safe to define this conditional KO as 'brain'-specific KO without explaining cell-type specificity.

[Page 5, line 9. "Extracellular K⁺ is taken up by astrocytes is quickly dissipated within a gap-junction connected astrocyte network." Reference is missing.]

We thank this reviewer to point this out. A paper that we cited in the following sentence (Wallraff et al., *J Neurosci.* 26:5438-5447, 2006) explains this. We combined two sentences to avoid confusion.

[Page 5, last sentence. It is not clear how this statement is related to K⁺ uptake by Kir4.1 channels.]

We are sorry for the confusion of this sentence. This sentence is just to explain that astrocyte gap-junction coupling is required for suppressing the neuronal network. To avoid the confusion, this sentence is omitted in the revised version, and the involvement of astrocyte network in neuronal network activity is explained in the sentence just before the deleted sentence.

[Figure 2D,F. Top panels - In order to avoid confusion and maintain consistency

with panel B (left), the plots here should also compare the relative free ubiquitin rather than depletion of free ubiquitin.]

We thank this reviewer for this advice. We changed the figure accordingly.

[Figure 3B, p.9. A lack of change in the protein level of GluA1 is not a sufficient evidence for ruling out the possible contribution of altered AMPA receptor activity to the reduced power of gamma oscillations. Synaptic surface level of AMPA receptors needs to be examined rather than the total receptor content, and more importantly, GluA2-containing AMPA receptors could be more relevant in influencing basal network activity. The authors should rephrase this statement.]

We agree with the reviewer. Our statement of aberrant AMPA receptor only by showing the total level of GluA1 is oversimplified. We have now changed "aberrant AMPA receptor" to "aberrant total GluA1 level, although the involvement of GluA2 is not excluded." to be precise. I hope this reviewer accept this change.

[p.13. "Cultured astrocytes were prepared from the cortex but not hippocampus because of homogeneous expression patterns of Cx43 and N4-2 throughout the cortex and hippocampus." The logic is not clear here. Perhaps the authors wish to justify the use of cortical astrocytes rather than hippocampal astrocytes, despite the fact that oscillation experiments have been performed in hippocampal slices - is this correct? The sentence needs to be clarified to convey the intention that there is an assumption here that cortical astrocytes should behave in a similar way as hippocampal astrocytes given the similar expression levels of Cx43 and N4-2 between the two brain regions.]

Sorry for the confusion. Indeed, we aimed to justify the usage of cortical astrocyte culture in this sentence. We changed the text in the revised version (page 14, lines 18 to 20).

[Figure 5 is disjointed from the flow of the manuscript. Does fluoxetine affect the I-V curve in control astrocytes? Does dynasore treatment in controls shift the I-V curve relative to mutant astrocytes or the mutant + dynasore condition? As it stands, the figure might be best included as supplementary information.]

We applied fluoxetine to the control astrocyte to study its impact on I-V curve. As shown in the revised Figures S2A and S2D, fluoxetine had almost no effect on the I-V

curve in control astrocytes. Unfortunately, dynasore application changed passive membrane properties; input resistance and resting membrane potential, indicating that dynasore is toxic for astrocytes. This could be explained potentially by dynasore's blocking all diamine-dependent endocytosis, by blocking dynamin-independent cholesterol homeostasis (Park et al., *JCS* 126: 5305-5312, 2013) or mTORC1 activity (Persaud et al., *JCS* 131: 2018, doi:10.1242/jcs.211755).

We feel that it is reasonable to include this result as a main figure in Figure 5 because the impact of Nedd4-2-dependent regulation of Kir4.1 is one of the main messages in our manuscript. We have another set of data of I-V curves in Figures 7K to 7M to support our conclusion from Figure 5.

[Figure 7N-P. It is not just the number of PV interneurons but it is their synaptic inputs and outputs that affect the CA3 circuit dynamics.]

We agree that synaptic connectivity of PV interneurons influences the circuit dynamics. We performed this experiment in Figure 7N-P for two reasons; (i) Cx43 plays crucial roles in tangential migration of inhibitory neurons (Elias et al., *J Neurosci.* 30: 7072-7077, 2010), and (ii) disturbed migration of inhibitory neurons might affect their cell numbers in the CA3 region, resulting in changes in synaptic connectivity. We believe that this dataset is important to demonstrate that Nedd4-2 AstKO has no developmental problem through increased Cx43 that could potentially influence the number of PV interneurons.

[Figure 8. Does fluoxetine or GAP26 cause a further increase in the power of gamma oscillations in control slices?]

We thank this reviewer for pointing this out. As shown in Figures S2E to S2H in the revised manuscript, fluoxetine application to the control slice had a minor impact on the power and frequency of γ -oscillation. Together with the result in Figures 8A to 8E, this result indicates that the function of Kir4.1 is upregulated in Nedd4-2 AstKO.

[Page 16, line 2. Connexins are not generally defined as ion channels as they also permeate larger molecules.]

Thank you for pointing this out. We removed 'ion' from the main text accordingly (page 17, line 3).

Reviewer #2 (Comments to the Authors (Required)):

[In this manuscript, Atlas and colleagues describe a mechanism through which E3 ubiquitin ligase Nedd4-2 in astrocytes regulate γ -oscillatory neuronal network activity. They show that Nedd4-2 controls the degradation of ion channels Kir4.1 and Connexin43 in astrocytes. Using a combination of mouse genetics, proteomics, in vitro and in vivo biochemical analyses and ubiquitination assays, the authors found that loss of Nedd4-2 results in a significant increase in the protein levels of both Kir4.1 and Connexin43. The presence of Nedd4-2 increases the levels of K63 ubiquitination on Kir4.1 and Connexin43, targeting these proteins for degradation via the endo-lysosomal pathway. Physiologically, loss of Nedd4-2 in astrocytes increases membrane conductance in a Kir4.1-dependent manner and increases gap junction coupling in vitro. Nedd4-2 astrocyte knockout animals show reduced gamma-oscillations in response to kainate treatment, which can be partially rescued with pharmacological inhibition of Kir4.1 or Connexin43. Mutations in Nedd4-2 are found in familial epilepsy, and the authors' data suggests that these mutations result in Nedd4-2 gain of function resulting in faulty ion buffering in astrocytes.]

In sum, this is an exciting study that potentially uncovers a novel molecular mechanism in astrocytes which is potentially relevant to epilepsy.]

We are all glad to know that this Reviewer finds our work exciting and that it has a potential to uncover novel mechanism in astrocytes.

[1. The manuscript starts out by using Nedd4-1 and Nedd4-2 double knockouts, including the initial screen which identified Kir4.1 and Cx43. The comparison is made to a CTL (control ? group) of unknown genotype(s) and backgrounds. The authors should clearly state which genotypes were used as controls.]

We thank this reviewer for pointing this out. We used *Nedd4-1^{fl/fl};Nedd4-2^{fl/fl}* mouse as the control for *Nedd4-1^{fl/fl};Nedd4-2^{fl/fl};EMX-Cre* mouse in proteome and electrophysiological studies. We explain this in lines 13 and 14 in page 9 in the revised manuscript. The background of all animals used in this study is C57BL/6N, as stated in the Animal Experiment in Methods section (5th and 6th lines from the bottom in page 25).

[2. Related to 1, there is also strong concern that the authors may have failed to completely knockout Nedd4-1 in their double KO model since there are faint bands in Figure 3B and F when probing for Nedd4-1 protein expression. Please explain.]

EMX-Cre driver line used in these Western blotting experiments expresses Cre recombinase in the offspring of radial glia cells specifically (Gorski et al., *J. Neurosci.*

22: 6309-6314, 2002). The faint bands from Nedd4-1 and Nedd4-2 double knockout samples in Figure 3B and 3F are likely from inhibitory neurons, blood cells, blood vessel cells, or microglia cells. In fact, roles of Nedd4-1 in T-cell (Yang et al., *Nat. Immunology* 9: 1356-63, 2008) and in endothelial cells have been reported (Xu et al., *Exp Cell Res.* 383: 111505, 2019). This is explained in the legend for Figure 3 (lines 4 to 6) in page 46.

[3. The authors use Eaat2 protein expression as an indirect indicator of astrocyte number. This is not an accurate analysis, and an actual cell count by staining with astrocyte markers in the CA3 is required to fully make the claim that astrocyte numbers did not change.]

We now include the result of quantitative immunohistochemistry using S100 β as an astrocytic marker in Figures 3L and 3M. This result indicates that astrocyte number is not changed in the absence of Nedd4-1 and Nedd4-2. We are hoping that our new dataset convinces this reviewer.

[4. There is a sharp difference in the power of γ -oscillations when comparing N4-2 KO (1G) and N4-2 astrocyte specific KO (Figure 8D and 8I), suggesting that there might be a non-astrocyte contribution to neuron network synchronicity i.e. a neuronal component this was not discussed at all. Please explain.]

To study the role of neuronal N4-1 and N4-2 in the maintenance of gamma-oscillation, we recorded the field potential from N4-1/2 neuron-specific conditional KO (nKO) mice. We used N4-1/2 nKO instead of N4-2 nKO simply because of the limited number of N4-2 nKO mice in our colonies. Results shown in Figures S1A to S1D demonstrate that neither of neuronal N4-1 or N4-2 is involved in the maintenance of gamma-oscillation. This result is described in Results (page 9, lines 14-16).

[5. The authors suggest that N4-1 compensates for N4-2 without any experimental proof. This may be OK to include in discussion as a possibility but should be omitted from the results.]

As requested, we deleted descriptions of the compensation of Nedd4-2-loss by Nedd4-1 in Results.

[6. The authors suggest that ubiquitination by N4-2 activates the lysosomal pathway, resulting in endocytosis of Kir4.1 and Cx43. However, they fail to explain for the seemingly additive effect on Cx43 protein expression (Figure 4H) when both lysosome activity and ubiquitination is blocked, which suggest that N4-2 mediated endocytosis is not the only factor regulating Cx43 cell surface expression.]

We agree with this reviewer that Ned4-2-mediated ubiquitination is not the only factor regulating Cx43 cell surface expression. In the bottom dot plot in Figure 4H (new Figure 4I), however, the impact of dynasore is more prominent in control astrocyte (1.76-fold; compare the first and second plots) than in N4-1/2 bDKO astrocytes (1.26-fold; compare the third and fourth plots). This result indicates that the effect of dynasore is mediated by N4-2, strongly suggesting that N4-2 plays a crucial role in endo-lysosomal pathway to downregulate Cx43. This argument is described in the 3rd line from the bottom in page 13 to the 3rd line in page 14. We hope this reviewer will accept our argument.

[7. Finally, there was only a partial rescue in the power of γ -oscillation when N4-2 KO astrocytes were treated with fluoxetine (Figure 8D). Since the authors argue that fluoxetine is a specific inhibitor of Kir4.1, this suggests that dysregulation of K⁺ buffering is not the only factor influencing neuronal network activity. Maybe Kir4.1 and CX43 work in paralele rather than in tandem. The authors should comment on this and potentially perform an experiment where both Cx43 and Kir4.1 are blocked.]

As this reviewer points out, it is likely that N4-2 plays roles by ubiquitinating two substrates in parallel because Kir4.1 and Cx43 localize in distinct compartments of astrocytes (i.e. tripartite synapse and GAP junction). We prepared for blocking both Kir4.1 and Cx43 in N4-2 AstKO in γ -oscillation recording. However, we are currently not allowed to perform invasive experiments such as injection of tamoxifen because our permission for invasive experiments were expired. We negotiated the local authority and found that it would take half a year to have a new permission. To avoid further delay of the submission of the manuscript, we explained this possibility in the Result (page 16, lines 4 to 9), as this reviewer suggests.

[Minor points

1. In Figure 4A, the authors should re-run the samples in the correct order for the figure so that there is no cut-and-paste involved.]

Samples in Figure 4A was run on the same SDS-PAGE gel. We made this point clear in the revised legend for Figure 4A (lines 28 and 29 on page 46). Splicing lanes is not ideal but is allowed according to JCB biowrites on November 15, 2015 in the link below. In the revised version, the result from experiments including the information same as the one in Figure 4A is presented as an uncut blot in Figure 4C. We hope this reviewer will waive this request.

<https://jcb-biowrites.rupress.org/figure-preparation/>

[2. In Figure 7D-G, where does the sample come from for Western blot? Is it from hippocampal lysate, cortical, or whole brain?]

We thank this reviewer for pointing out the lack of the details in this Western blot. We used cortical lysate for this quantitative Western blotting. The source of samples is now stated as “cortical lysate” in the figure legend for Figure 7 [(D) and (E to G)] in the revised manuscript.

[3. In Figure 7, the authors indicate that 5% of TdTomato+ cells are not positive for S100beta, indicating that a cell type other than astrocytes is undergoing Cre-mediated recombination. Since the authors are using a tamoxifen dosing timeline which differs from previously published studies (in which tamoxifen was administered in the adult), they should determine whether any NeuN+ (and in particular PV+) cells are expressing TdTomato.]

According to this reviewer’s comment, we studied the expression of tdTomato in NeuN+ or PV+ cells at the CA3 region of the hippocampus. As shown in Figure 1 for the reviewer in the next page, we did not detect any tdTomato expression in either cell groups. Unfortunately, it is very difficult to include this figure in the manuscript, even in the supplemental data, because of the limitation of the figure numbers. I hope this reviewer agrees that we state this results in the main text as below (lines 12 to 14 in page 15); “We confirmed that there were no tdTom-expressing cells stained with neuronal marker NeuN (650 cells in 4 mice) or with the inhibitory neuronal marker Parvalbmine (27 cells in 4 mice).”

[4. The authors make an incorrect statement in the discussion section on p.19. Cx43 in astrocytes pairs with Cx47 on oligodendrocytes, not with Cx32 as the authors have stated.]

We corrected this error, as requested by the reviewer (page 20, line 13). We thank this reviewer for pointing this out.

[5. In the methods section, please clarify the method used for Quantitative Western blotting. The description of methods for the In vitro Ubiquitination Assay indicated that fluorescent secondary antibodies and the Odyssey Imaging System were used, but it is unclear for the blots in other figures. Was this same method of quantification used?]

We used fluorescent secondary antibodies and the Odyssey Imaging System for all quantitative Western blotting experiments. This is explained in the “Quantitative Western blotting” in the Methods section in the revised manuscript (page 27, the 4th from the bottom to the bottom lines).

Reviewer #3 (Comments to the Authors (Required)):

[In the manuscript entitled 'Physiological and pathophysiological homeostasis of astroglial channel proteins by *Nedd4-2*', Altas and colleagues have demonstrated

that a ubiquitin E3 ligase Nedd4-2 ubiquitinates and downregulates expression of Kir4.1 and Cx43 in astrocytes, leading to increase in the membrane ion permeability and in gap junction connectivity. In general, the manuscript is well written but there are some important issues, which need to be solved to support their conclusion.]

We are very happy to know that this reviewer thinks that our manuscript is well written.

Major points:

[1. Figure 1. as used in Figure 3M-R, it is critical to show gamma-Oscillations using the N4-1bKO samples to support their conclusion.]

We thank this reviewer for pointing this out. We agree that this is an important negative control. We include this new result as Figures S1E-S1H in the revised version.

[2. in Figure 2, what is the reason why E271A is not tested for its activity in vitro? It would be important to test this mutant to strengthen the statement of authors.

We agree with this reviewer; it is important to test if E271A mutation would boost the E3 ligase activity. We have tried to purify the recombinant protein from E.coli. However, we could not achieve a good yield; this point mutation seem to make this E3 ligase unstable in E.coli. Instead of *in vitro* assay, we showed increased E3 ligase activity of E271A point mutant in cell-based ubiquitination assay using HEK293FT cells (Figure 4C). We hope that the result of this assay, where Kir4.1 and Cx43 are used as substrates, would convince this reviewer.

Based on the method section, used samples are not denatured before IP. This is absolutely critical to conclude if Cx43 is ubiquitinated or not, especially because HA-IP, HA-Blt samples seem not to show modification of HA-Cx43.]

We are sorry for confusing the reviewer regarding the method description for cell-based ubiquitination assay. We actually did denature the proteins in HEK293FT cell lysate using 1% SDS-containing buffer before the IP. Please note that we incubated samples at 65 °C but not at 100 °C to avoid aggregation of recombinant transmembrane substrate proteins. The details of these experiments are now documented in the revised method section (page 31, lines 1 to 14).

[3. Following up the above point #2, for all the in cell ubiquitination assays, it is

essential to denature once the lysates before IP. (after denaturing, one needs to adjust the buffer suited for IP)]

All in cell ubiquitination assays, which we call *in vivo* or cell-based ubiquitination assay in the manuscript, were performed in the original manuscript, as this reviewer suggests. We also cited a paper where this protocol is described (Staub et al., *EMBO J.* 16: 6325-6336, 1997) in the method section for the purpose of clarification of the method.

[4. Figure 4B: Regarding authors' description as 'We interpret this data as compensatory homeostatic regulation of Cx43-HA by K63-ubiquitination-dependent lysosomal degradation.', treatment of cells with Bafilomycin A1 would clarify this point. Since it is rather simple experiments and will help authors' conclusion, it is important to test BafA1-treated samples.]

To address this, we applied another lysosomal inhibitor chloroquine instead of BafA1 because BafA1 killed HEK293 cells in our experiments. As shown in Figure 2 for the reviewer, chloroquine increased the level of Cx43-HA when EGFP-Nedd4-2 was mildly co-expressed. However, even after repeating the experiments 5 times, we failed

to show the significance with the ANOVA test. Thus, we removed our statement 'We interpret this data as compensatory homeostatic regulation of Cx43-HA by K63-ubiquitination-dependent lysosomal degradation.' from the main text.

[5. Figure 3H: Especially because evidence of Dynamin inhibition in these samples are not shown, and because a more recent study has shown about Dynamin-independent roles of Dynasore (PMID: 29150487 DOI:10.1242/jcs.211755), it is better to rephrase the following sentence; To study whether N4-2 promotes the

degradation of Kir4.1 and Cx43 via the endolysosomal pathway, clathrin-mediated endocytosis was blocked with dynasore, a specific blocker of dynamin (Macia et al., 2006) in primary astrocyte cultures (Figure 4H).]

In the paper that Reviewer#3 suggests, Dynasore could be a potential suppressor of mTORC1. If Kir4.1 and Cx43 expressions were targets of PI3K/mTORC1 pathway, Dynasore treatment would suppress their expression rather than enhance them as shown in Figure 4H. However, it remains possible that a target of PI3K/mTORC1 pathway might reduce expressions of Kir4.1 and Cx43 indirectly. We changed the text from 'a specific blocker of dynamin (Macia et al., 2006)' to 'a blocker of dynamin (Macia et al., 2006)' at the 3rd line from the bottom in page 13.

[6. it is very difficult to understand what kind of samples are used if it is written only as 'samples'. For example, it is unclear what were analysed in Figure 3I and J from the figure legends nor the text. Thus, it is very hard for readers to understand if the conclusion made by authors such as 'controls their surface expression (page 16 line 3)' is supported by the provided data or not. Please fix.]

We thank this reviewer for pointing out that we omitted the identities of samples in the original version of our manuscript by mistake. Samples used for Figures 3I and 3J, which are Figure 3J and 3K in the revised version, are cortical lysates. Ones for Figures 7D to 7G are also cortical lysates. The identities of samples are now described in each figure legend of the revised manuscript.

[7. in the Discussion section: page 16, line 2: The authors state that a single E3 ligase ubiquitinates XXXX, but based on the data in Figure 4C and D, and as authors interpreted these data sets as 'probably by other E3s involved', it is very confusing where this statement came from.]

We apologize our causing confusion with the statement in the Discussion. We changed the Discussion from 'by which a single E3 ligase, N4-2, coordinately ubiquitinates' to 'by which N4-2 ubiquitinates' to avoid the confusion.

[8. Figure 2 and 10: N4-2 mutants S233L or H515P forms more free ubiquitin chains in vitro but no evidence provided if Kir4.1 and Connexin43 are more ubiquitinated. Without examining the ubiquitination status of the substrates by these mutants at least in vitro, it is not possible to conclude if the effect on the

protein expression levels of the substrates are direct or indirect. Even though Fig 10 is just a schema, it is suggested to perform such experiments to test one of the main points of this study.]

We thank this reviewer for this point. As requested, we performed the suggested experiments and present the results in the revised Figure 4C. Of note, we changed the description of one of mutations from H515P to H536P when we use mouse Nedd4-2 in the experiment. The amino acid number 515 in human NEDD4-2 corresponds to 536 aa in mouse Nedd4-2.

Minor points:

[1. in Figure 2, to obtain better ideas where the three human mutations of N4-2 are localized, it is recommended to add a similar schematic as Fig 4E and indicate the mutation sites.]

We thank this reviewer for the advice. The suggested, the scheme is now included as Figure 2C in the revised manuscript.

[2. in the Method section, Quantitative Western blotting: It will be very helpful to comment one phrase that the Odyssey system was used as this is critical info (although which is properly indicated in the Hsia et al paper).]

We comment this in the “Quantitative Western blotting” part of the Method section (page 27, the 5th line from the bottom to the bottom lines).

[3. page 9, line 8-9: the statement 'N4-1 and N4-2 generate K63-linked polyUbi chains (Maspero et al., 2013), which are involved in the endocytosis and lysosomal degradation of transmembrane proteins (Acconcia et al., 2009).' is not entirely correct. As Acconcia et al wrote in their review, only some specific cases like AMSH-dependent EGFR degradation is controlled by K63 chains. Please rephrase.]

We are sorry that our description was confusing. We have now changed this sentence, as requested (page 10, line 11-13).

[4. page 9, Data not shown. It is highly recommended to add the data possibly as a supplementary figure.]

As shown in the Figure 3D, the level of Nedd4-2 was almost the same in cortical and hippocampal lysates.

[5. page 17, line 3-5, please indicate at least citations for this statement ' Nevertheless, the major consequences of loss-of function of a specific E3 in a given cell type are often due to aberrant ubiquitination of only a few substrates, indicating a substantial cell-type specificity and selectivity of E3-substrate interactions.' since this concept seems not to be generalized in the research field.]

We included new citations as examples (page 18, line 5). Hengstermann et al, showed that ubiquitination of p53 by E6AP is crucial for cell growth. Tokunaga et al., showed LUBAC-dependent ubiquitination of NEMO is of particular importance in NF-kappaB signaling.

[6. page 17, regarding the statement 'The present study demonstrates that a dominant function of N4-2 in the adult brain is to SPECIFICALLY and SELECTIVELY ubiquitinate and downregulate Kir4.1 and Cx43 in astrocytes.': based on the data in 3E, there are more targets shown in double KO samples of N4-1 and 2. The chosen targets as Kir4.1 and Cx43 in this study are indeed the N4-2 substrates, however, it is unclear if the above statement can be made based on the shown data in this study.]

We are sorry for our confusing statement. We used these two terms in this sentence to emphasize that the ubiquitination of Kir4.1 and Cx43 is of importance in this particular context, regulation of neuronal network activity. We changed this sentence to avoid this confusion in the revised version.

[7. please fix the qRT-PCR method section from the protocol style to the method style.]

We changed the style for the method of qRT-PCR as suggested by this reviewer (starting from page 28, line 7).

November 18, 2021

Re: JCB manuscript #201902050R-A

Dr. Hiroshi Kawabe
Max Planck Institute of Experimental Medicine
Department of Molecular Neurobiology
Hermann Rein Str. 3D
Göttingen 37075
Germany

Dear Dr. Kawabe,

I am attaching the evaluations of two reviewers of your revised submission to this journal. Happily, the reviewers evaluate very positively the improvements in data and description made in your revision. While we are not quite in position to accept the work, I anticipate that I can make a rapid editorial decision when you have had a chance to consider and make appropriate revisions to your study.

I have read the manuscript together with comments of the reviewers. For simplicity, I will give you mine and the journal's assessments of each and their importance for inclusion in your revision.

The original reviewer #2 has 6 major points:

The first addresses statistical and data presentation issues. I agree with this reviewer that these issues should be addressed following his/her recommendations. This happily should not require additional experiments.

The second point concerns the issue of mixing cortical biochemical with hippocampal physiological data. If you now have hippocampal data, it would greatly strengthen the manuscript to include it. If not, I would just ask that you make the assumption of equivalency as clearly as possible in both the experimental description and the final discussion.

The third point seems reasonable to me. It also does not seem to require extensive effort on your part. I think it is reasonable for you to provide the impact on levels of the Cx43 and Kir4.1 proteins (this is the only new data that I think must be included).

Regarding point 4, I do not think it is essential to include studies on the Nedd4-2 catalytically dead mutant.

Regarding point 5, I would request that you include a bar graph with the quantification requested by the reviewers as an addition to Figure 4.

Regarding point 6, while I agree with the assessment of the reviewers, I do not think it is essential for you to do the slice experiments recommended by this reviewer.

To summarize, both the reviewers and I believe that your work is very close to being appropriate for publication. I would encourage you to perform the single experiment (item 3 of original reviewer #2), make the appropriate textual changes and return the manuscript as quickly as possible.

Please include a letter describing your changes and a version of the manuscript in which textual changes are marked with either some form of highlighting or 'tracked changes'. I hope to make a final decision very quickly upon resubmission.

Sincerely yours,

Louis F. Reichardt
For the JCB

Tim Spencer, PhD
Executive Editor
Journal of Cell Biology

Reviewer #2 (Comments to the Authors (Required)):

In this revised manuscript, Altas et al report a new study investigating the role of Nedd4-1 and Nedd4-2 in regulating astrocyte membrane proteins, Cx43 and Kir4.1. Using a combination of genetic dissection, biochemical assays and slice and in vivo electrophysiological recordings, the authors find that Nedd4-2 is the predominant E3-ubiquitin ligase that regulates the

abundance of Cx43 and Kir4.1 and brain-specific knockout of Nedd4-2 causes a significant increase in these two proteins via K63 ubiquitination. Further, this study reveals that epilepsy mutants of Nedd4-2 disrupt gamma oscillations in the hippocampus of mice, which can be partially rescued by blocking either Cx43 or Kir4.1 pharmacologically. Together this study is well-controlled and of high interest to the community especially given the epilepsy link. The authors have responded well to the previous reviews. Some additional comments are added below which should be taken into consideration to improve statistical analyses and data presentation. There are also some proposed experiments, but these are suggestions and authors could address these points by textual clarifications and discussion.

Reviewer's comments

1. While there is sufficient power from the number of recorded slices (n=20-30), the data can be better interpreted if animal number was included, and the plots (Fig 1C, D, G and H) were colored to show which points belong to which animal to account for possible variability. Based on the spread of the data (Fig 1C and 1G), it is possible that the decrease in frequency is driven by a portion of the animals. Nested t-test is a more stringent statistical test to use and would give more confidence in the claim that decreased power of γ -oscillations is largely driven by N4-2 if the p value holds up. This critique is also true of Fig S1C/G
2. γ -oscillation recordings were performed in the hippocampus (CA3) but quantitative mass spectrometry was done on cortical lysates. Considering regional heterogeneity of hippocampal and cortical astrocytes, the fact that both the cortex and hippocampus express similar levels of Nedd4-2 (Fig4D) is not sufficient proof that Kir4.1 and Cx43 are Nedd4-2 substrates in the hippocampus. We suggest repeating Fig 3G/3S with hippocampal lysates to demonstrate upregulation of Kir4.1 and Cx43 protein levels to increase confidence in this claim.
3. The authors show very nicely that loss of Nedd4-2 results in increased Cx43 and Kir4.1. They also show that the gain of function patient mutants of Nedd4-2 increase the ubiquitination of Cx43 and Kir4.1. For completeness, the authors should show that Cx43 and Kir4.1 proteins are downregulated when the gain of function Nedd4-2 mutants are expressed.
4. In this study, the authors study 3 epilepsy-linked mutants of Nedd4-2. Interestingly, these mutants block ubiquitination of the substrates despite not being in the known catalytically active HECT domain. Further, these mutants are not present in any annotated Nedd4-2 domain. The authors should compare their data from these mutants to the Nedd4-2 catalytically dead mutant in which the conserved cysteine in the HECT domain is mutated. (See Gao et al, Mol. Cell. 2009, PMID: 19917253).
5. The authors claim that Figure 4G shows ~2-fold more Kir4.1 HA and Cx43-HA bound to purified GST-tagged N4-2 than to GST-N4-1. However, there is no quantification to support this. The authors should either add the quantification or revise their claim.
6. The images in Figure 6 do not look like primary astrocytes. This experiment could easily be done in slices. An ex vivo preparation will be more informative as astrocytes cultured alone are starkly different than those in vivo.

Minor points of correction to note:

1. GAP26 as an inhibitor of Cx43 should be introduced first for clarity, instead of first mentioning its molecular weight relative to fluoxetine.
2. Spelling error in pg 9, line 15 "neuron-speCCific N4-1/N4-2 KO mice"
3. Grammar in pg10, line 17 "after determining that the cortex expresses N4-1 and N4-2 at the levels comparable to"

Reviewer #3 (Comments to the Authors (Required)):

The authors addressed all the suggested points by the reviewer. The reviewer has no further concerns.

Point-by-Point Response to Reviewer's Comments (Reviewer's Comments Bold)

We have been grateful for the critical but thoughtful comments of the Reviewer and the Editor. As explained below, we made a serious effort to follow their advice, and we believe that our manuscript has been improved significantly as a consequence, e.g. by a new set of experiments and by editing the manuscript text. Below, we explain how we approached the issues raised by Reviewer 2 in a point-by-point response.

Reviewer 2

1. While there is sufficient power from the number of recorded slices (n=20-30), the data can be better interpreted if animal number was included, and the plots (Fig 1C, D, G and H) were colored to show which points belong to which animal to account for possible variability. Based on the spread of the data (Fig 1C and 1G), it is possible that the decrease in frequency is driven by a portion of the animals. Nested t-test is a more stringent statistical test to use and would give more confidence in the claim that decreased power of γ -oscillations is largely driven by N4-2 if the p value holds up. This critique is also true of Fig S1C/G

As suggested, we included the animal numbers in figure legends and labeled the plots in Fig. 1C, 1D, 1G, and 1H as well as in Fig. S1C, S1D, S1G, and S1H accordingly. Changing the statistical test from Student's t-test to nested t-test changed the p-value from 0.014 to 0.076 for the data shown in Fig. 1G. As the Reviewer points out, the nested t-test is more stringent than Student t-test. As indicated in Fig. A below (for the Reviewer), data for control mouse 4 (N4-2CTL) have a higher average than those of all other control mice, and data from KO mouse 1 (N4-2 bKO) have the lowest average in the N4-2 bKO group. When excluding any one of these datasets, the Student's t-test yields $p=0.032$ and $p=0.038$, respectively, indicating that the significant difference between the groups is not only caused by a subset of animals. In the revised manuscript, we now describe our corresponding results as follows: "The average power of γ -oscillatory activity was significantly reduced in brain-specific conditional N4-1/N4-2 KO mice (*Nedd4-1^{fl/fl};Nedd4-2^{fl/fl};EMX-Cre*; N4-1/2 bDKO) (Gorski et al., 2002; Kawabe et al., 2010; Kimura et al., 2011) or showed a strong trend toward a significant reduction ($p=0.076$) in brain-specific N4-2 KO mice (*Nedd4-2^{fl/fl};EMX-Cre*; N4-2 bKO) as compared to their corresponding controls (*Nedd4-1^{fl/fl};Nedd4-2^{fl/fl}*; N4-1/2 CTL and *Nedd4-2^{fl/fl}*; N4-2 CTL respectively) when tested with a stringent statistical test (i.e. nested test) (Figure 1).

Student's t-test indicated that N4-2 bKOs show a significantly reduced power of γ -oscillatory activity as compared to controls (p= 0.014)."

Figure A (for the Reviewer) γ -Oscillations in acute slices from Nedd4-2 brain-specific KO mice. Data are shown for individual mice. N4-2 CTL mouse 4 showed the highest and N4-2 bKO mouse 1 the lowest average γ -oscillatory power in the corresponding groups.

2. γ -oscillation recordings were performed in the hippocampus (CA3) but quantitative mass spectrometry was done on cortical lysates. Considering regional heterogeneity of hippocampal and cortical astrocytes, the fact that both the cortex and hippocampus express similar levels of Nedd4-2 (Fig4D) is not sufficient proof that Kir4.1 and Cx43 are Nedd4-2 substrates in the hippocampus. We suggest repeating Fig 3G/3S with hippocampal lysates to demonstrate upregulation of Kir4.1 and Cx43 protein levels to increase confidence in this claim.

We appreciated this comment. However, we still believe that quantitative Western blotting using the cortical lysate supports our conclusion given the ratios of hippocampal and cortical expressions of Nedd4-2, Kir4.1, and Cx43 are similar as indicated in the links below in Allen Brain Atlas.

<https://mouse.brain-map.org/experiment/show?id=74047904>

<https://mouse.brain-map.org/experiment/show?id=69735700>

<https://mouse.brain-map.org/experiment/show?id=79556642>

Instead of performing the experiment, we explained clearly that these experiments were performed using cortical lysates (page 11 line 1) and the rationale of our experimental design and the results (page 10 line 19, page 17 lines 14 to 18). We are hoping that this reviewer agrees with us.

3. The authors show very nicely that loss of Nedd4-2 results in increased Cx43 and Kir4.1. They also show that the gain of function patient mutants of Nedd4-2 increase the ubiquitination of Cx43 and Kir4.1. For completeness, the authors should show that Cx43 and Kir4.1 proteins are downregulated when the gain of function Nedd4-2 mutants are expressed.

We thank the Reviewer for his*her advice. We expressed HA-tagged Kir4.1 or Cx43 in HEK293FT cells together with wild type or point mutant Nedd4-2. As shown in Fig. B, overexpression of recombinant Nedd4-2 reduces the levels of Kir4.1 and Cx43 as compared to mock transfected cells. Unexpectedly, this effect was comparable to that of overexpression of point mutants of Nedd4-2. Given that the levels of ubiquitination of substrates are increased in HEK293FT cells expressing the mutant Nedd4-2 (Fig. 4C), we interpreted our findings to indicate that the strongly increased amount of Kir4.1 and Cx43 overwhelms the lysosomal degradation system in HEK293FT cells so that no further degradation can be achieved. A related possibility

Figure B (for the Reviewer) HEK293FT cells were transfected with expression vectors of Myc-Nedd4-2, HA-substrate with IRES-EGFP sequence. (A) Western blotting using the lysate of HEK293FT cells. The red channel indicates the signal from HA-tagged Kir4.1 (lanes 1 to 5) and Cx43 (lanes 6 to 10). Note the level of HA-substrates was reduced upon overexpression of WT Nedd4-2 (compare 1st and 5th lanes, and 6th and 10th lanes), while mutation of Nedd4-2 shows no effects on HA-substrate levels. (B) Statistical analyses of the results in (A). N = 7 independent experiments.

to explain this negative result is that overexpression of Nedd4-2 causes strongly increased ubiquitination of multiple endogenous substrates (e.g. ENaC), causing a 'traffic jam' from endosome to lysosome. In order to circumvent such problems, we tried to optimize the expression levels of recombinant protein by changing the promoter and the amount of plasmids used for the transfection. However, we could not detect mutation-dependent reductions of Kir4.1 and Cx43 levels. This experiment seemed simple originally, but turned out to be very difficult. We would like this Reviewer to waive the requirement to pursue this line of inquiry.

4. In this study, the authors study 3 epilepsy-linked mutants of Nedd4-2. Interestingly, these mutants block ubiquitination of the substrates despite not being in the known catalytically active HECT domain. Further, these mutants are not present in any annotated Nedd4-2 domain. The authors should compare their data from these mutants to the Nedd4-2 catalytically dead mutant in which the conserved cysteine in the HECT domain is mutated. (See Gao et al, Mol. Cell. 2009, PMID: 19917253).

As this Reviewer suggests, an additional experiment using the loss of function mutant of Nedd4-2 would complete this experiment. However, it is extremely well established that the point mutant of the aforementioned conserved cysteine entirely eliminates the E3 ligase activity. Therefore, further comparisons with this mutant seem to us beyond the scope of our manuscript and unlikely to yield information that would further bolster our conclusions. We would like this Reviewer to waive the requirement to pursue this line of inquiry.

5. The authors claim that Figure 4G shows ~2-fold more Kir4.1 HA and Cx43-HA bound to purified GST-tagged N4-2 than to GST-N4-1. However, there is no quantification to support this. The authors should either add the quantification or revise their claim.

We apologize for our incomplete description in this regard. In the revised Fig. 4G, we now included bar diagrams for this quantification.

6. The images in Figure 6 do not look like primary astrocytes. This experiment could easily be done in slices. An ex vivo preparation will be more informative as astrocytes cultured alone are starkly different than those in vivo.

Our manuscript demonstrates the cell biological and biochemical processes regulated by Nedd4-2. Cultured cells are ideal for this purpose. Other datasets *in situ* and *in vivo* flank this analysis and support the findings made in cultured cells. Besides, the absence of the neuron in the culture system guarantees the cell autonomous roles of astrocytic Nedd4-2. We would like this reviewer to waive this request.

Minor points of correction to note:

1. GAP26 as an inhibitor of Cx43 should be introduced first for clarity, instead of first mentioning its molecular weight relative to fluoxetine.

In the revised manuscript, we have addressed this issue (page 16, lines 5 to 6).

2. Spelling error in pg 9, line 15 "neuron-speCCific N4-1/N4-2 KO mice"

This was corrected in the revised manuscript.

3. Grammar in pg10, line 17 "after determining that the cortex expresses N4-1 and N4-2 at the levels comparable to"

We are sorry for this grammatical error. Our point was to state that the expression level of N4-2 is almost same in cortex and hippocampus. We changed the manuscript accordingly.

September 22, 2023

RE: JCB Manuscript #201902050RR

Prof. Hiroshi Kawabe
Gunma University
3-39-22 Showa-machi
Maebashi, Gunma 371-8511
Japan

Dear Prof. Kawabe:

Thank you for submitting your revised manuscript entitled "Physiological and pathophysiological homeostasis of astroglial channel proteins by Nedd4-2". We would be happy to publish your paper in JCB pending final revisions necessary to meet our formatting guidelines (see details below).

A. MANUSCRIPT ORGANIZATION AND FORMATTING:

1) Text limits: Character count for Articles and Tools is < 40,000, not including spaces. Count includes the abstract, introduction, results, discussion, and acknowledgments. Count does not include title page, materials and methods, figure legends, references, tables, or supplemental legends. You are currently below the limit but please bear it in mind when revising.

2) Figures limits: Articles and Tools may have up to 10 main text figures.

3) Figure formatting: Scale bars must be present on all microscopy images, including inset magnifications. Molecular weight or nucleic acid size markers must be included on all gel electrophoresis.

4) Statistical analysis: Error bars on graphic representations of numerical data must be clearly described in the figure legend. The number of independent data points (n) represented in a graph must be indicated in the legend. Statistical methods should be explained in full in the materials and methods. For figures presenting pooled data the statistical measure should be defined in the figure legends. Please also be sure to indicate the statistical tests used in each of your experiments (both in the figure legend itself and in a separate methods section) as well as the parameters of the test (for example, if you ran a t-test, please indicate if it was one- or two-sided, etc.).

****Also, since you used parametric tests in your study (e.g. t-tests, ANOVA, etc.), you should have first determined whether the data was normally distributed before selecting that test. In the stats section of the methods, please indicate how you tested for normality. If you did not test for normality, you must state something to the effect that "Data distribution was assumed to be normal but this was not formally tested."****

5) Title: The title should be less than 100 characters including spaces and should be concise but accessible to a general readership. We feel that your current title is a bit too vague to be appreciated by all audiences. Therefore we suggest the following title: "Nedd4-2 modulates neuronal network activity via regulation of Kir4.1 and Connexin43".

6) Materials and methods: Should be comprehensive and not simply reference a previous publication for details on how an experiment was performed. Please provide full descriptions (at least in brief) in the text for readers who may not have access to referenced manuscripts. The text should not refer to methods "...as previously described."

****Please note that we cannot accommodate a table within the Methods section. Therefore, we recommend you convert the antibodies table into a separate "Supplementary Table" and simply cite it in the text of the methods (e.g. "See Supplementary Table 1 for a list of all antibodies used in this study").****

7) Please be sure to provide the sequences for all of your primers/oligos and RNAi constructs in the materials and methods. You must also indicate in the methods the source, species, and catalog numbers (where appropriate) for all of your antibodies.

8) Microscope image acquisition: The following information must be provided about the acquisition and processing of images:

- a. Make and model of microscope
- b. Type, magnification, and numerical aperture of the objective lenses
- c. Temperature

- d. imaging medium
- e. Fluorochromes
- f. Camera make and model
- g. Acquisition software
- h. Any software used for image processing subsequent to data acquisition. Please include details and types of operations involved (e.g., type of deconvolution, 3D reconstitutions, surface or volume rendering, gamma adjustments, etc.).

10) Supplemental materials: There are strict limits on the allowable amount of supplemental data. Articles/Tools may have up to 5 supplemental figures. At the moment, you are below this limit but please bear it in mind when revising. Please also note that tables, like figures, should be provided as individual, editable files. A summary of all supplemental material (that is, in addition to the supplementary figure legends) should appear at the end of the Materials and methods section. Please see any recent JCB paper for an example of this.

11) eTOC summary: A ~40-50 word summary that describes the context and significance of the findings for a general readership should be included on the title page.

****We realize that you have already provided a summary but please note that the statement should be written in the present tense and refer to the work in the third person. It should contain "First author name(s) et al..." to match our preferred style.****

13) A separate author contribution section is required following the Acknowledgments in all research manuscripts. All authors should be mentioned and designated by their first and middle initials and full surnames. We encourage use of the CRediT nomenclature (<https://casrai.org/credit/>).

14) ORCID IDs: ORCID IDs are unique identifiers allowing researchers to create a record of their various scholarly contributions in a single place. Please note that ORCID IDs are now ***required*** for all authors. At resubmission of your final files, please be sure to provide your ORCID ID and those of all co-authors.

15) Journal of Cell Biology now requires a data availability statement for all research article submissions. These statements will be published in the article directly above the Acknowledgments. The statement should address all data underlying the research presented in the manuscript. Please visit the JCB instructions for authors for guidelines and examples of statements at (<https://rupress.org/jcb/pages/editorial-policies#data-availability-statement>).

B. FINAL FILES:

****It is JCB policy that if requested, original data images must be made available to the editors. Failure to provide original images upon request will result in unavoidable delays in publication. Please ensure that you have access to all original data images prior to final submission.****

****The license to publish form must be signed before your manuscript can be sent to production. A link to the electronic license to publish form will be sent to the corresponding author only. Please take a moment to check your funder requirements before choosing the appropriate license.****

Additionally, JCB encourages authors to submit a short video summary of their work. These videos are intended to convey the main messages of the study to a non-specialist, scientific audience. Think of them as an extended version of your abstract, or a short poster presentation. We encourage first authors to present the results to increase their visibility. The videos will be shared

on social media to promote your work. For more detailed guidelines and tips on preparing your video, please visit <https://rupress.org/jcb/pages/submission-guidelines#videoSummaries>.

Thank you for your attention to these final processing requirements. Please revise and format the manuscript and upload materials within 7-14 days. If complications arising from measures taken to prevent the spread of COVID-19 will prevent you from meeting this deadline (e.g. if you cannot retrieve necessary files from your laboratory, etc.), please let us know and we can work with you to determine a suitable revision period.

Please contact the journal office with any questions, cellbio@rockefeller.edu.

Thank you for this interesting contribution, we look forward to publishing your paper in Journal of Cell Biology.

Sincerely,

Louis Reichardt, PhD
Monitoring Editor
Journal of Cell Biology

Tim Spencer, PhD
Executive Editor
Journal of Cell Biology